# SHOT2STORY: A NEW BENCHMARK FOR COMPREHENSIVE UNDERSTANDING OF MULTI-SHOT VIDEOS

**Mingfei Han**[1,2,3,5†*]**, Linjie Yang**[1†]**, Xiaojun Chang**[3,4]**, Lina Yao**[5]**, Heng Wang**[1]
[1]Bytedance Inc.   [2]ReLER Lab, AAII, UTS   [3]Department of Computer Vision, MBZUAI
[4]University of Science and Technology of China   [5]Data61, CSIRO
https://github.com/bytedance/Shot2Story

## ABSTRACT

A short clip of video may contain progression of multiple events and an interesting story line. A human need to capture both the event in every shot and associate them together to understand the story behind it. In this work, we present a new multi-shot video understanding benchmark Shot2Story with detailed shot-level captions, comprehensive video summaries and question-answering pairs. To facilitate better semantic understanding of videos, we provide captions for both visual signals and human narrations. We design several distinct tasks including single-shot video captioning, multi-shot video summarization, and multi-shot video question answering. Preliminary experiments show some challenges to generate a long and comprehensive video summary for multi-shot videos. Nevertheless, the generated imperfect summaries can already achieve competitive performance on existing video understanding tasks such as video question-answering, promoting an under-explored setting of video understanding with detailed summaries.

## 1 INTRODUCTION

Video captioning is a long-standing video understanding task to facilitate open-world video analysis with the help of human-annotated captions. Since a video may contain multiple events, dense captioning benchmarks (Ego4D (Grauman et al., 2022), YouCook2 (Zhou et al., 2018), ActivityNet-Caps (Krishna et al., 2017)) capture information of multiple events in videos ranging from 3-20 minutes. However, even within seconds, more than one event often occurs in daily videos such as news broadcasts, tutorial videos, and movies. Specifically, shot transition, which is a common technique to transfer from one event to another, or to switch the viewpoint of a single event, happens less than every 4s for average English movies after 2010 (Cutting et al., 2011). Although some existing captioning benchmarks (Xu et al., 2016; Krishna et al., 2017; Zhou et al., 2018) already use multi-shot videos, they often annotate the captions in a coarse-grained manner, either providing a holistic caption or asking annotators to subjectively choose the boundary of each event. To better accommodate the multi-shot formation of videos, we believe a new video benchmark with rich textual descriptions based on video shots is favored in the research community.

On the other hand, multi-shot videos are often accompanied by rich narrations that relate to the different events happening in the video. A model needs to capture both the visual and audio signals to understand the underlying story. Specifically, narrations may contain key information that cannot be inferred from pure visual information only. See Figure 1, without the narration, a viewer is unable to capture the relationship between the man's action and the avocado product in the first shot.

In this work, we propose a new benchmark Shot2Story for audio-visual understanding of multi-shot videos. We collect a dataset of 42,958 short videos where the average number of shots in each video is 4.4. For each video shot, we annotate a detailed textual description for the video frames and another textual description for the human speech. We also leverage a state-of-the-art large language model (LLM) GPT-4 (OpenAI) to generate a long textual video summary from the annotated clip descriptions, which are further verified by human annotators. The summary includes additional details such as transitions of different shots, progression of multiple events, and mapping of the subject identities in different scenes. An example of one annotated video is shown in Figure 1.

---

*Work was done during an internship at Bytedance. † Equal contribution.

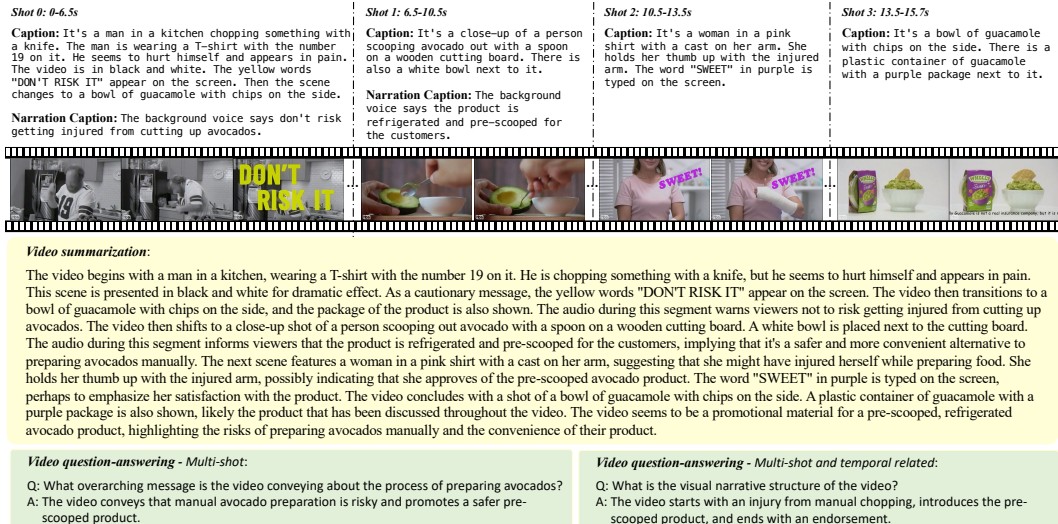

Figure 1: An annotated example of our Shot2Story with sing-shot visual captions and narration captions. Moreover, we provide coherent and reasonable video summaries, and question-answering pairs to facilitate comprehensive understanding of multi-shot videos.

To benchmark the advances of multi-modal video understanding, we designed several distinctive tasks using our dataset, including single-shot video captioning, multi-shot video summarization, and multi-shot video question answering. We design and implement several baseline models using a frozen vision encoder and an LLM, by prompting the LLM with frame tokens and ASR (Automatic Speech Recognition) text. Through extensive experiments, we show that: (1) the ASR text is critical to joint understanding of visual and audio content, (2) processing the video as a whole without the shot structure degenerates the model's capacity of understanding the multi-shot video, (3) the summarization model trained on our multi-shot summaries can be used on the proposed multi-shot QA benchmark and generalized to other datasets with longer durations (ActivityNet(Krishna et al., 2017)) and out-of-domain topics (MSRVTT(Xu et al., 2016)), validating the quality of our annotated summaries. Without any bells and whistles, we attain competitive results on zero-shot video question-answering by converting the problem into pure text-based QA with the generated video summaries.

## 2 THE SHOT2STORY BENCHMARK

Our new benchmark Shot2Story contains 42,958 videos. The length of each video is ranging from 10s to 40s. We first use an off-the-shelf shot detection method TransNetV2 (Souček & Lokoč, 2020) to split each video into shots. For each video shot, we annotate captions for both visual and audio information. Then we further annotate video summaries based on the annotated shot captions. Figure 2 shows an overview of our dataset with some key statistics.

### 2.1 DATA PREPARATION

We source videos for our dataset from the public video benchmark HD-VILA-100M (Xue et al., 2022). It offers a large collection of narrative videos, comprising 3M YouTube videos segmented into 100M clips, each about 13 seconds long. We choose this data source for its concise yet complex multi-shot formats, diverse topics, and abundant ASR content. Since we prefer videos with both rich visual and ASR information, we design several filtering techniques to exclude those videos with either low visual-ASR correlation or static visual content.

We start with keeping video clips with durations between 10 to 40 seconds, since we observe that the majority of the video clips from HD-VILA-100M fall in this range. Then we remove videos with more than 8 shots due to the heavy annotation cost. We also notice that the video segments with too many shots in HD-VILA-100M tend to be slideshows or image collages that deviates from our focuses. Further, to harvest videos with rich visual-ASR correlations, we set up a metric between

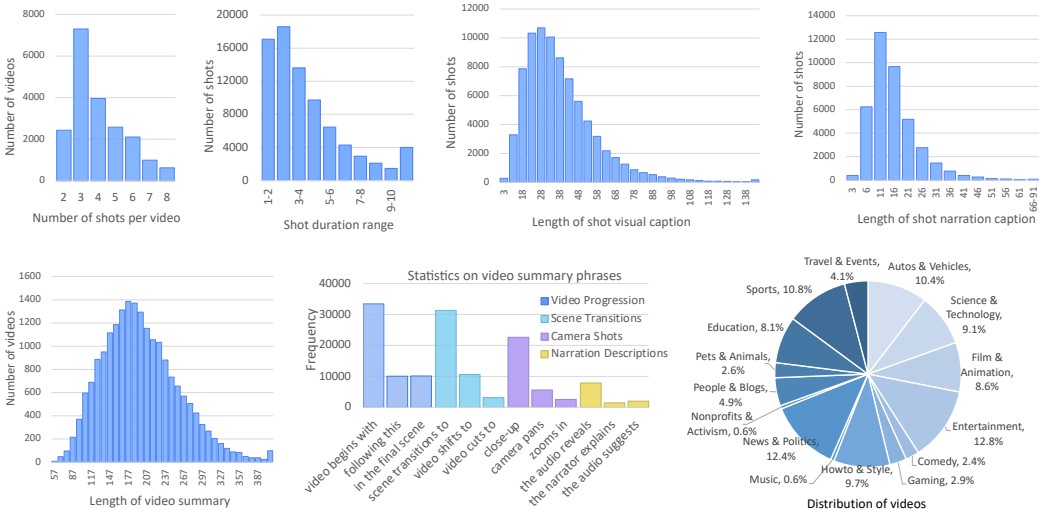

Figure 2: Statistics of Shot2Story . Our dataset features detailed visual captions and narration captions, and video summaries, highlighting video progressions, transitions, camera cuts and narration descriptions, with statistics of frequent expressions depicted in the figure.

video shots and ASR texts. Specifically, we uniformly sample 4 video frames for each shot and obtain the cosine similarity score between the video shot embedding and the text embedding using CLIP (Radford et al., 2021) encoders. We only keep the videos containing at least one shot that is visually correlated to ASR with a threshold of 0.25. Next, in order to obtain videos with diverse shot contents, we set up an inter-shot metric to filter out the videos with similar adjacent shots. We compute the cosine similarities between embeddings of adjacent shots and keep the videos with all inter-shot similarity scores smaller than 0.9. Finally, to further remove the videos with static contents, we adopt an intensity-based scene change detector in PySceneDetect[1] with a low threshold of 11 on our segmented shots. If the filter is unable to detect scene changes at this low threshold, it is conceivable that the shot contains static contents. We only keep the video clips in which all shots contain no static content based on our filtering method.

As a result, from a total of 2.1M sampled video clips from HD-VILA-100M, we obtain $42,958$ video clips that meet our quality standard. The number of shots in each video is from 2 to 8. These videos are then shared with our annotators for further annotations.

## 2.2 ANNOTATION OF SINGLE-SHOT CAPTIONS

After using TransNetV2 to divide the target videos into video shots, we ask annotators to annotate both visual-only captions and audio-related captions for each shot. We split these two caption annotation to facilitate separate modeling of these two types of information source. For visual-only captions, we require annotators to describe the major subjects and events in the video. Since it is an open-world setting, the videos can be quite diverse and hard to describe. In order to reduce the difficulties of annotating a caption from scratch, we generate an initial video caption using MiniGPT-4 (Zhu et al., 2023) by sampling 4 image frames from the video clip and prompting the model using below prompt.

*###Human:Frame1</Img >Frame2</Img>Frame3</Img>Frame4</Img>Please describe this video. Do not include details that you are not sure of. For example, if there is text in the image, do not include the content of the text if they are not clearly shown. ###Assistant:*

Although MiniGPT-4 is originally designed for image understanding, empirically it is able to generate captions for short video clips, both comprehensively and reasonably. It is able to describe different subjects including persons, animals, food, tools, and virtual objects like animated characters. Annotators are first instructed to correct any errors in the original captions. The mistakes include incorrect descriptions of the object categories, attributes, actions, facial expressions, etc. Also, there might be some subjective descriptions generated by MiniGPT-4 such as emotion and atmosphere. We ask annotators to remove all these subjective descriptions. We then ask annotators to supplement the

---

[1]https://www.scenedetect.com/

Table 1: High-level comparison of our dataset to previous ones. The summary length of ActivityNet and YouCook2 are combined length of captions in one video. M and G denote manual and generated.

| Dataset | Annotation | Multi-shot Video | Multi-event Descriptions | Detailed Summary | Summary Length | #Videos | Avg. Duration |
|---|---|---|---|---|---|---|---|
| MSRVTT (Xu et al., 2016) | M | ✓ | ✗ | ✗ | - | 10K | 15s |
| ActivityNet Caps (Krishna et al., 2017) | M | ✓ | ✓ | ✗ | 52.4 | 20K | 3min |
| VideoStorytelling (Li et al., 2019) | M | ✓ | ✓ | ✓ | 162.6 | 105 | 12.5min |
| Ego4D (Grauman et al., 2022) | M | ✗ | ✓ | ✗ | - | 10K | 23min |
| YouCook2 (Zhou et al., 2018) | M | ✓ | ✓ | ✗ | 67.8 | 2K | 6min |
| VAST (Chen et al., 2024) | G | ✓ | ✗ | ✓ | 32.4 | 27M | 5∼30s |
| Shot2Story | M+G | ✓ | ✓ | ✓ | 218.3 | 43K | 17.1s |

information about the major subjects, actions, and backgrounds present in the video. The goal is for the resulting captions to accurately capture the key elements of each video shot. Statistics shows over 80% of single-shot visual captions are manually corrected. For narration captions, annotators should watch the video and interpret the audio content that visually correlates into descriptive narration, including information sources. For example: "According to the woman in white, the room is not very clean." All narration captions are manually drafted from scratch. An example of this annotation process is shown in Appendix A.1, where the annotator corrects the caption from "standing in front of the car" to "getting close to the car", and adding a missing detail of "a close-up shot of the front". In this way, we find the annotation speed significantly faster (∼ 3×) compared to writing a caption from scratch. On the other hand, we find the captions generated this way has more coherent style and tend to cover more details of the video.

In contrast to the traditional video captioning benchmarks (Xu et al., 2016; Krishna et al., 2017; Zhou et al., 2018), we also annotate narration captions in addition to the visual-only captions. Different from existing audio captioning benchmarks (Gemmeke et al., 2017), we focus more on human speeches rather than acoustic events. Annotators are required to associate the human speech with the video content and summarize the main idea of the speech. We require annotators to describe the source of the speech using visual information. For example, if someone is talking, the annotators need to describe which person in the video is talking. If the human speech refers to some object in the video, the annotator is required to describe which object in the video the speaker is referring to. Note that the speaker identity and reference of visual objects are critical information for understanding a video that cannot be trivially obtained using existing algorithms. There are existing research on speaker identification (Kim et al., 2021) and visual grounding (Anne Hendricks et al., 2017; Zhou et al., 2019), but they only work well on constraint scenarios.

## 2.3    ANNOTATION OF VIDEO SUMMARY

To create video summaries with annotated video-shot captions, we leverage an LLM-based approach. Specifically, we form a text prompt with incorporating all shot captions and ASR text included, and uses GPT-4 (OpenAI) to generate a cohesive summary. The text prompt we use is shown in Appendix A.2. The quality is assured through further review and correction by our annotators.

We prompt GPT-4 to produce coherent, fluent text summaries with transition expressions such as "the video begins", "following this", and "in the final scene" to connect video-shot descriptions. The generated annotations also encompass a higher-level understanding of shots, using key phrases such as "scene shifts back" and "returns to the scene" to denote recurring scenes across shots. Notably, GPT-4 often identifies and links the same subjects across scenes without relying on explicit re-identification models. It draws on descriptive and attributive text from shot captions like "a newsroom" or "a man wearing a black suit" to infer scene or subject identity. To ensure quality, annotators carefully review and correct any inconsistencies in scene or subject references within summaries. Since our shot-level captions for generating video summaries are manually checked and annotated, the initial video summaries merely have factual errors, with exceptions for some subject identity and scene mismatches. We require annotators to pay more attention to these errors and ensure holistic summary is accurate and comprehensive. Statistics show that over 40% summaries are manually corrected.

Despite the rigorous verification process, the reliance on automated generation introduces certain inaccuracies and biases. Common pitfalls in the generated summaries include the omission of minor yet contextually important details and a bias towards emphasizing more prominent actions or objects,

potentially overlooking less conspicuous elements. This is partly due to our dataset predominantly featuring human-centric activities, as a result of our video filtering process that selects videos rich in visually related audio information and sourcing from HDVILA Xue et al. (2022), which primarily curates content related to human activities. Consequently, our annotations tend to highlight salient events and large-scale objects essential to the video's storyline, mentioning smaller objects only when they directly contribute to the narrative.

## 2.4 ANNOTATION OF QUESTION-ANSWERING PAIRS

We annotate the question-answering pairs on videos in validation and testing splits. To construct this benchmark, we begin with the human-annotated video summaries from Section 2.3, which is detailed in video content. We then prompt GPT-4 (Achiam et al., 2023) to generate candidate question-answer pairs in three predefined categories: temporal-related (*e.g.*, *Does the woman appearing at the end of the video wear any accessories?*), multi-shot holistic understanding (*e.g.*, *What is the overarching theme of the video?*), and audio content related (*e.g.*, *Which specific car model does the background voice mention, and what visual features confirm its identity?*). The text prompt we use is shown in Appendix B.3.

The quality is then assessed through further automatic filtering and manual checks. Annotators are instructed to verify each question-answer pair against the video content and discard any with mistakes. Simultaneously, they are asked to categorize the questions, where a single question might fall into multiple categories, facilitating evaluating different aspects of multishot understanding *i.e.*, understanding sequences of events and actions (temporal related), integrating information across multiple shots (holistic understanding), correlating audio content with visual elements (audio related), and others (directly discarded). After this verification, annotators carefully review the quality of the questions to ensure they solely correlate with the video content and that the answers are not revealed in the question texts. This step includes a thorough manual check to address potential mistakes and biases from the initial verification. Through these sequential annotation stages, we ensure high-quality annotations. The process starts with detailed manual single-shot captioning, followed by careful verification and correction at each stage. Even with LLMs introduced to reduce workload, the human-involved process and thorough procedures keep the resulting annotations well-aligned with human labeling.

Subsequently, to optimize clarity and reduce unnecessary information in the QA pairs, we remove QA pairs with questions exceeding 28 words or answers exceeding 20 words. Then, we employ Vicuna-13B (Chiang et al., 2023) to attempt answering the questions without video context and discard those answerable without accessing the videos. Finally, the pairs are tested against popular methods such as Video-LLaVA (Lin et al., 2023), LLaMA-VID (Li et al., 2023e), Video-ChatGPT (Maaz et al., 2023), and VideoChat2 (Li et al., 2023c). Questions correctly answered by more than two models are excluded to guarantee that our dataset poses a substantial challenge. Finally, we obtain 4905 QA pairs for validation and 6465 QA pairs for testing set.

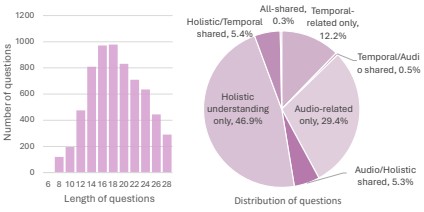

Figure 3: Distribution of multi-shot video QA benchmark. Questions from different categories overlap. *All-shared* means questions fall under all three categories.

## 2.5 COMPARISON TO EXISTING BENCHMARKS

Compared to existing video description datasets, our dataset is more challenging due to the explicit modeling of the multi-shot nature of web videos. Our textual description includes both shot-level captions and video-level summaries, combining visual and audio understanding, which provides a unique test bed for multi-modal video understanding. Table 1 shows a high-level comparison of our new dataset with existing video captioning benchmarks. Most existing video captioning benchmarks, such as MSRVTT (Xu et al., 2016), YouCook2 (Zhou et al., 2018) and ActivityNet Caps (Krishna et al., 2017), also use multi-shot videos as annotation source, but they either annotate a holistic caption for the video (MSRVTT) or ask annotators to decide the boundary of different events. In our study, we observe that video shots naturally create a sequence of related events, motivating us to annotate distinct captions for each shot. Ego4D (Grauman et al., 2022) only annotates dense visual captions but not audio captions for relatively long egocentric videos. Video Storytelling (Li et al.,

Table 2: Performance of models on video shot captioning using different modalities, following the settings of VAST(Chen et al., 2024). The models are fine-tuned on Shot2Story video shot captions. V, A and S are abbreviated for vision, audio and subtitle (ASR text) respectively.

| Model | FT Modality | B4 | M | R | C |
|---|---|---|---|---|---|
| VAST | V+S | 10.7 | 16.1 | 30.3 | 33.8 |
| VAST | V+A+S | 10.7 | 16.1 | 30.4 | 34.0 |
| MiniGPT4-C | V | 9.2 | 14.7 | 27.9 | 25.1 |
| MiniGPT4-C | V+S | 11.8 | 16.7 | 30.1 | 35.9 |
| VideoChat2-C | V | 8.8 | 16.1 | 27.9 | 23.7 |
| VideoChat2-C | V+S | **13.8** | **18.7** | **32.1** | **43.9** |

Table 3: Performance of models on video summarization. The models are fine-tuned on Shot2Story video summaries. V and S are abbreviated for vision and subtitle (ASR text) respectively.

| Model | FT Modality | B4 | M | R | C |
|---|---|---|---|---|---|
| Video-ChatGPT w/o ASR (Maaz et al., 2023) | V | 4.8 | 17.3 | 21.3 | 1.5 |
| Video-ChatGPT (Maaz et al., 2023) | V+S | 3.6 | 17.8 | 19.7 | 1.0 |
| MiniGPT4-SUM-holistic | V+S | 7.8 | 16.9 | 23.4 | 2.8 |
| MiniGPT4-SUM-shot w/o ASR | V | 10.4 | 18.5 | 25.8 | 4.8 |
| MiniGPT4-SUM-shot | V+S | 12.4 | 19.7 | 27.6 | 7.6 |
| VideoChat2-SUM-shot | V+S | **12.7** | **19.8** | **28.3** | **9.0** |

2019) is a small-scale dataset with annotations of multiple events in a videos and provides a summary of the video by concatenating all captions.

A recent work VAST (Chen et al., 2024) feeds generated video and audio captions into an LLM to generate video summary. However, it processes multi-shot video as a whole and lacks the granularity of the events in different shots. Moreover, VAST directly uses predicted captions without any human verification, leading to potentially noisy and biased summaries towards the captioning models. Our dataset stands out from VAST with its accurately annotated visual and narration shot captions. Although our video summary is also generated using an LLM, it is further verified by annotators to make sure there is no hallucinated details from the LLM. Our dataset has an average length of 218.3 words for the video summary, which is much longer than existing benchmarks, and is longer than the combined length of captions in one video in ActivityNet and YouCook2.

Furthermore, our Shot2Story-QA introduces unique and complex challenges through its emphasis on shot transitions and multi-event progression, setting it apart from benchmarks like MSRVTT-QA Xu et al. (2017) and ActivityNet-QA Yu et al. (2019). For instance, unlike existing benchmarks Xu et al. (2017); Yu et al. (2019) that assess understanding at a single time point, *e.g.*, "Who do three judges talk to?" (MSRVTT-QA, in Figure 19), or general inquiries like "What is the person in the video doing?" (ActivityNet-QA, in Figure 20), Shot2Story-QA includes "Temporal-related" and "Multi-shot Holistic Understanding" questions. A temporal-related question, such as "What is the man's immediate action after handling the skewer?" shown in Figure 21, requires models to comprehend the sequence and progression of events, linking consecutive actions meaningfully. Similarly, multi-event progression questions like "How does the setting change from the start to the end of the video?" necessitate understanding multiple concurrent events within their temporal context. This provides a more rigorous and nuanced evaluation framework for temporal understanding.

## 3 TASKS AND EXPERIMENTS

### 3.1 BASIC SETTINGS

For all the tasks described in this section, we follow the same training/validation/test split. Specifically, the number of videos for training, validation, and test set are 36951, 1982 and 4025, respectively. We resize the frames to $224 \times 224$. We adapt two popular VLMs to accomodate our tasks: MiniGPT-4 (Zhu et al., 2023) and VideoChat2 (Li et al., 2023c). For MiniGPT-4, we employ ViT-G/14 (Fang et al., 2022) and Q-Former (Li et al., 2023a) as visual encoder, and Vicuna v0-7B (Chiang et al., 2023) as the language model. We load pretrained Q-Former and MLP from MiniGPT-4 (Zhu et al., 2023). In training, we update only Q-Former and MLP parameters, keeping the ViT and LLM frozen.

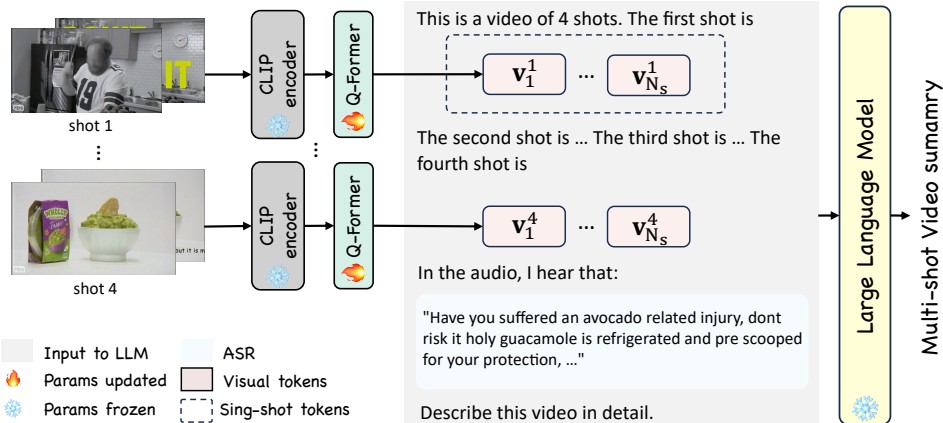

Figure 4: Model structure for multi-shot video summarization model SUM-shot. We arrange visual tokens sequentially for each single shot and in a multi-shot format to encapsulate multi-shot information. Additionally, ASR text is incorporated for audio-visual video summarization.

For VideoChat2, we employ UMT-L(Li et al., 2023d) as backbone and load pretrained Q-Former and MLP from VideoChat2 (Li et al., 2023c). During training, we adopt LoRA(Hu et al., 2021) and AdamW (Loshchilov & Hutter, 2017) with a learning rate of 8e-5. We train both models for 10 epochs with a batch size of 128 for single-shot video captioning. We finetune our video summarization models on the single-shot captioning models with a batch size of 32. For captioning and summarization, we evaluate the models using BLEU@4 (Papineni et al., 2002) (B), METEOR (Denkowski & Lavie, 2014) (M), ROUGE (Lin, 2004) (R), and CIDEr (Vedantam et al., 2015) (C).

## 3.2 SINGLE-SHOT VIDEO CAPTIONING

This task involves generating descriptions for individual video shots, where the target description is a concatenation of the visual-only and narration caption for a video shot. This task requires a joint understanding of visual and speech information. Specifically, we adopt a similar structure as we generate pseudo captions for data annotation in Section 2.2. First, we sample $N_s$ frames from a video shot, encode them using a fixed vision encoder, then feed the encoded features to a Q-Former to produce visual tokens. Further, we combine the visual tokens and an optional ASR text into a unified LLM prompt to facilitate both visual and narration understanding. We adapt the framework of MiniGPT-4 (Zhu et al., 2023) and VideoChat2 (Li et al., 2023c), with the two models denoted as MiniGPT4-C and VideoChat2-C for brevity. We compare with baseline model VAST (Chen et al., 2024), which is able to processes audio, vision, and subtitle inputs simultaneously.

The results are shown in Table 2. Benefiting from extensive pretraining, VAST achieves 34 on C and 30 on R, comparable to MiniGPT4-C. Incorporating an additional audio modality results in a negligible performance difference, indicating that audio content only contributes marginally given the ASR text as input on our benchmark. We then assess variants of our models, MiniGPT4-C and VideoChat2-C, with and without the additional ASR text. It shows that including ASR texts significantly enhances performance across all metrics, with a notable boost in R and C, highlighting the relevance of audio content to our video captions. Furthermore, VideoChat2-C, featuring a superior visual backbone and extensive video pretraining, consistently outperforms MiniGPT4-C. This superiority highlights the importance of advanced visual backbone and video pretraining, confirming our benchmark's robustness. However, despite these advances, the results also indicate room for improvement. Figure 5 (a) showcases output examples of our model's single-shot video captioning, detailing visual elements and audio content effectively, capturing actions like "gesturing to explain her fear" and secondary elements such as "a red stuffed doll next to her".

## 3.3 MULTI-SHOT VIDEO SUMMARIZATION

Multi-shot video summarization is a new task that is distinct from existing video description tasks. It requires the model to understand the shot structure of the given video and to provide a coherently paragraph to describe the progression of events in the different shots. Due to the complexity of this task, we adopt GPT-V (Achiam et al., 2023) to generate a supplementary training set, with video summaries for another 90K videos, sampled in the same approach as described in Section 2.1. Please

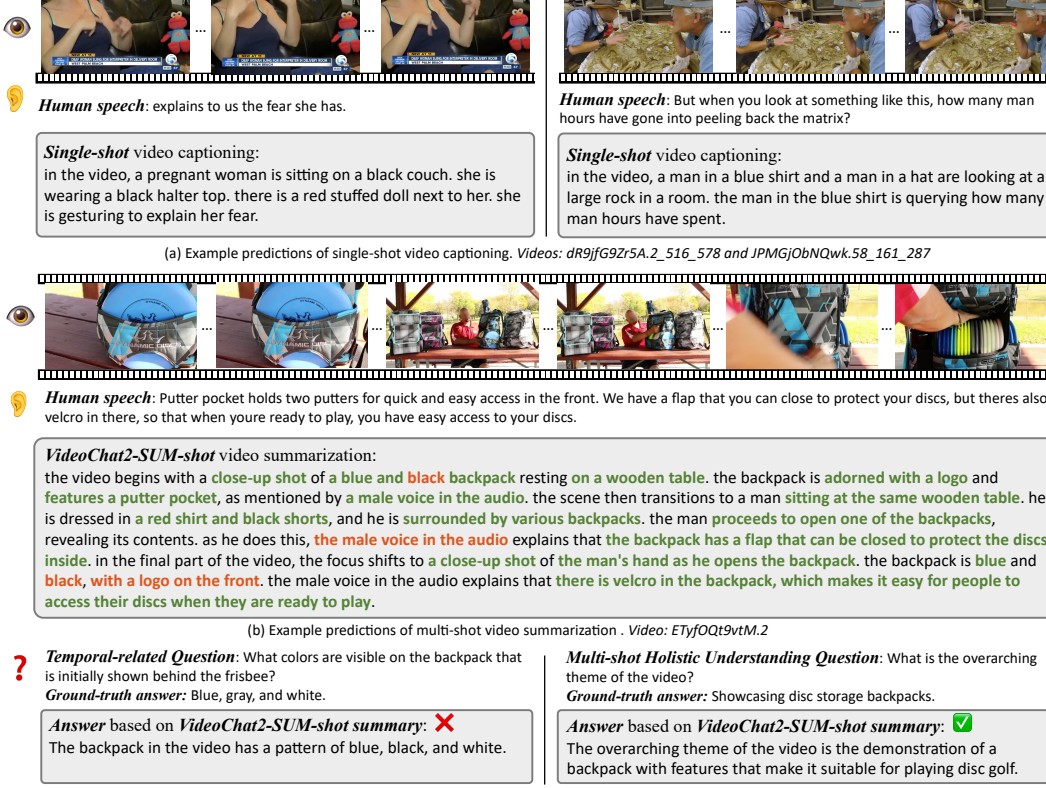

(a) Example predictions of single-shot video captioning. *Videos: dR9jfG9Zr5A.2_516_578 and JPMGjObNQwk.58_161_287*

(b) Example predictions of multi-shot video summarization . *Video: ETyfOQt9vtM.2*

(c) Example questions and predicted answers of multi-shot video. *Video: ETyfOQt9vtM.2*

Figure 5: Example predictions of our models. (a) shows single-shot video captioning results of VideoChat2-C, which incorporates audio and visual content correctly (b) shows multi-shot video summarization of VideoChat2-SUM-shot, with accurate descriptions in green and errors in red, illustrating the model's ability to narrate event sequences (c) shows two sample questions of the video in (b). The answers are based on the generated summary of VideoChat2-SUM-shot.

check the annotation prompt and data samples in Appendix B.5. First, we finetune an existing video caption model Video-ChatGPT (Maaz et al., 2023) by instruction-tuning it on our video summary data, with and without the additional ASR text input. Then, we experiment with three different architecture designs based on MiniGPT4. The first model MiniGPT4-SUM-holistic uses a similar pipeline as MiniGPT4-C. We uniformly sample 16 frames from the full video clip and prompt the LLM with frame tokens and ASR text. The second model MiniGPT4-SUM-shot w/o ASR, neglecting ASR input, uses a more refined framework by sampling 4 frames in each video shot and prompting the LLM with frame tokens from different shots, as is shown in Figure 4. The third model, MiniGPT4-SUM-shot further incorporates ASR text input as an additional input. Further, we replace the backbone of MiniGPT4-SUM-shot with the more advanced VideoChat2 model, resulting in the VideoChat2-SUM-shot model variant. Compared to SUM-shot, SUM-holistic does not have explicit shot information and relies on the LLM to parse the video shots using the provided frame features.

Table 3 shows the results of the models. It is shown that MiniGPT4-SUM-holistic is worse than MiniGPT4-SUM-shot, showing the importance of the shot structure in predicting a video summary matching the transition of shots. MiniGPT4-SUM-shot w/o ASR underperforms compared to MiniGPT4-SUM-shot and outperforms MiniGPT4-SUM-holistic, highlighting the significance of both audio information and shot structure in multi-shot video understanding. Compared VideoChat2-SUM-shot and MiniGPT4-SUM-shot, the former model achieves the best performance, indicating the benefit of advanced vision backbone and video pretraining. Video-ChatGPT obtains much worse performance comparing to our models, potentially due to their weakness in processing multiple scenes and building the correlation between visual frames and ASR texts. It directly encodes the whole video into a sequence of tokens, potentially losing significant frame details and essential correlation between ASR and visual frames, while ours directly feed frames tokens into the LLM without compressing them.

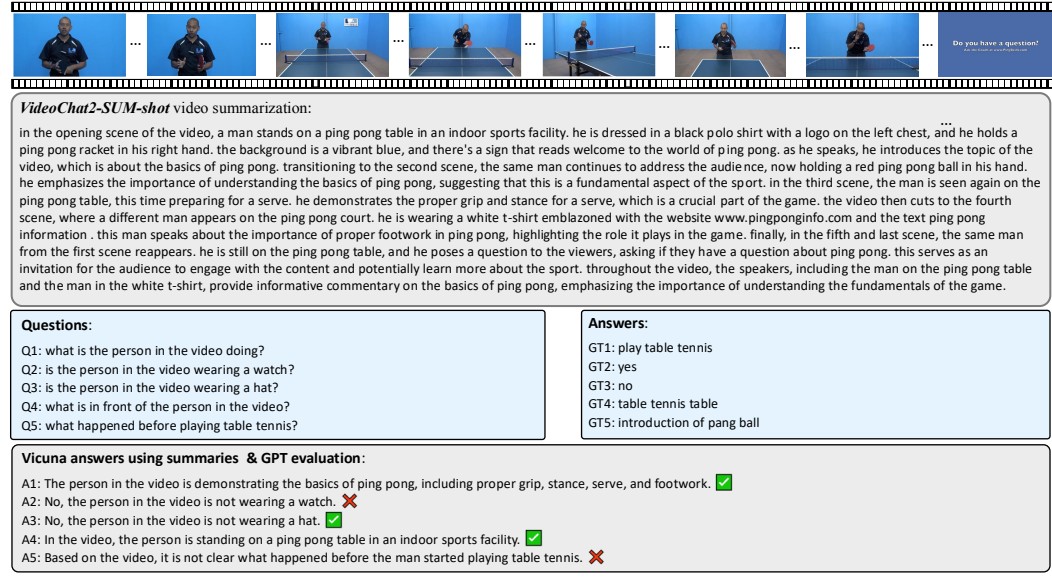

ActivityNet-QA case: v_mZYWfmsYQPA.mp4 *duration: 104 seconds*

Figure 6: Example predictions of our model on zero-shot question answering an ActivityNet-QA video. More questions and explanations can be found in Appendix C.2.

Figure 5 (b) showcases predictive capabilities of our VideoChat2-SUM-shot model. The model adeptly narrates event sequences with appropriate emphasis. For instance, it details the backpack's colour and location, and rationalizes the item in the beginning shot, *i.e.*, "putter pocket", aligning with the ASR with "by a male voice in the audio". However, some predictions that marked in red are erroneous, such as the incorrect "black" colour and the non-existent "with a logo on the front" in the ending shot. These inaccuracies likely stem from the LLM's tendency to "hallucinate" plausible yet non-factual details. Despite these errors, the model demonstrates proficiency in generating consistent and nuanced summaries, highlighting our model's potential and the challenges our dataset presents.

## 3.4 VIDEO QUESTION-ANSWERING WITH VIDEO SUMMARY

Generated video summaries are supposed to be grounded and detailed, covering rich elements like event progression, holistic topics and audio elements, making them suitable for other vision tasks such as video question-answering. Existing work (Guo et al., 2023; Zhang et al., 2023) uses image or video frame captions as input to an LLM to generate question responses. However, little work has been done for the capacity of video summaries. We directly apply our video summarization model on video QA benchmarks, *i.e.* MSRVTT-QA (Xu et al., 2017), ActivityNet-QA (Yu et al., 2019) and our Shot2Story-QA.

Specifically, we first split the testing videos into video shots, and then feed the videos into our SUM-shot models. The generated summaries and the associated questions are then fed into a Vicuna model to derive the answers with the prompt shown in Appendix B.4.1. Note there is no adaptation or finetuning conducted for the Vicuna model. Since the original answers in the QA benchmarks are very short and the responses generated by LLM tend to be long sentences, we leverage the `gpt-3.5-turbo` model to generate a binary decision on whether the answer is correct, following Video-ChatGPT (Maaz et al., 2023).

**Zero-shot video question-answering.** As shown in Table 4, our results with VideoChat2-SUM-shot surpass 5 out of 6 existing video-VLMs on MSRVTT-QA and 4 out of 6 existing models on ActivityNet-QA. Furthermore, our results are comparable to the SOTA performance on MSRVTT-QA with Video-LLaVA (Lin et al., 2023). Note that these models require extensive instruction-tuning data to learn to directly generate answers from visual features and the text prompt, whereas our model bypasses instruction tuning by distilling the video information into a summary. Our model also follows the zero-shot QA setting since the model only uses Shot2Story as training data. Note that MSRVTT contains a large portion of videos with out-of-domain topics such as TV shows

Table 4: Performance on video question answering on MSRVTT-QA and ActivityNet-QA. IT means whether the model uses video-text instruction tuning data. All methods follow the zero-shot manner.

| Model | Training Data | IT | QA Input | MSRVTT QA | ActivityNet QA |
|---|---|---|---|---|---|
| VideoChat (Li et al., 2023b) | Cap.+QA | ✓ | V+T | 45.0 | 26.5 |
| Video-ChatGPT (Maaz et al., 2023) | Cap.+QA | ✓ | V+T | 49.3 | 35.2 |
| MovieChat (Song et al., 2023) | Cap.+QA | ✓ | V+T | 52.7 | 45.7 |
| LLaMA-VID (Li et al., 2023e) | Cap.+QA | ✓ | V+T | 57.7 | 47.4 |
| VideoChat2 (Li et al., 2023c) | Cap.+QA | ✓ | V+T | 54.1 | **49.1** |
| Video-LLaVA (Lin et al., 2023) | Cap.+QA | ✓ | V+T | **59.2** | 45.3 |
| MiniGPT4-SUM-shot | Summary | ✗ | T | 57.7 | 45.6 |
| VideoChat2-SUM-shot | Summary | ✗ | T | 58.5 | 47.1 |

Table 5: Benchmark on Shot2Story-QA. IT means usage of video-text instruction tuning data. Summary, Cap. and QA denote video summary, captions and question-answering pairs.

| Model | Training data | IT | QA Input | Temporal related | Holistic understanding | Audio related | Overall |
|---|---|---|---|---|---|---|---|
| LLaMA-VID (Li et al., 2023e) | Cap.+QA | ✓ | V+T | 7.9 | 9.7 | 11.4 | 9.7 |
| Video-ChatGPT (Maaz et al., 2023) | Cap.+QA | ✓ | V+T | 13.1 | 15.5 | 14.3 | 14.2 |
| VideoChat2 (Li et al., 2023c) | Cap.+QA | ✓ | V+T | 15.1 | 15.4 | 13 | 14.5 |
| Video-LLaVA (Lin et al., 2023) | Cap.+QA | ✓ | V+T | 16.4 | 14.8 | 11.7 | 14.3 |
| MiniGPT4-SUM-shot | Summary | ✗ | T | 28.9 | 31.9 | 36.7 | 32.5 |
| VideoChat2-SUM-shot | Summary | ✗ | T | 36.1 | 41.5 | 43.8 | 40.5 |

(e.g., Figure 19) and food, while ActivityNet has much longer videos than our training videos (e.g., Figure 6. This validates the robustness and transferability of our model across different topics and longer videos. This surprisingly good result indicates that a comprehensive and detailed video summary is a high-quality abstraction of the video, facilitating a wide range of tasks including video QA and video-based conversation. Moreover, our model achieves strong results on ActivityNet-QA, which predominantly consists of single-shot long videos, validating that models trained with multi-shot videos can effectively generalize to single-shot videos.

**Multi-shot video question-answering.** As shown in Table 5, we benchmark existing and our proposed video summary models on Shot2Story-QA. Specifically, four popular video-VLMs are compared, *i.e.*, Video-ChatGPT (Maaz et al., 2023), LLaMA-VID (Li et al., 2023e), VideoChat2 (Li et al., 2023c) and Video-LLaVA (Lin et al., 2023). The predicted summaries of MiniGPT4-SUM-shot and VideoChat2-SUM-shot are used to tackle the QA task with the same configuration as zero-shot VQA. Accuracies on temporal-related, multi-shot holistic understanding and audio-related are reported, with the "overall" metric showing the average score from these three sub-tasks. The current video-VLMs present unsatisfying results, potentially due to two factors: (1) The current models does not have audio or ASR as input, lacking capacity with audio-related understanding. (2) Current models do not have training data with detailed descriptions based on multi-shot videos, weakening their performance on holistic understanding and temporal modeling. For our models, VideoChat2-SUM-shot achieves an overall score of 40.5, surpassing the compared models and MiniGPT4-SUM-shot on all three subtasks. This performance underscores the benefits of video pretraining and the advanced visual backbone of VideoChat2. These baseline results highlight the complexities and demanding nature of our Shot2Story-QA task. We show some example predictions in Figure 5(c). Please refer to Appendix C.2 for more qualitative results.

## 4 CONCLUSION

In this work, we present Shot2Story, a large-scale benchmark for comprehensive multi-shot video understanding. We provide detailed shot-level captions for both visual signals and human narrations. Furthermore, we provide comprehensive video summaries based on shot-level captions and design a challenging video question-answering benchmark for multi-shot video understanding. With the rich and diverse descriptions, our benchmark serves as a playground for future multi-modal video understanding models, ready to be extended for a range of other video understanding tasks, such as visual grounding and video-based conversation.

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
