# A ANNOTATION PROCESS

In this section, the annotation process for Shot2Story is detailed, including single-shot caption annotation in Sec. A.1, GPT-4 summary generation prompts in Sec. A.2, human correction of summaries in Sec. A.3, and annotation of question-answering pairs in Sec. A.4.

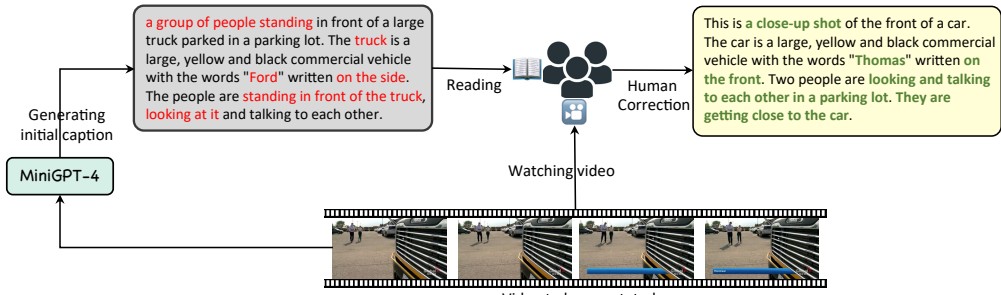

Figure 7: Human annotation process of sing-shot video captions. Texts in bold green represent correct content, while those in red indicate errors. Please find more explanations in Sec. A.1.

## A.1 HUMAN ANNOTATION OF SINGLE-SHOT CAPTIONS

Our single-shot video caption annotation process, described in Sec. 2.2, is a two-phase approach designed for high-quality and style-consistent captions. This procedure also accelerates the annotation process by ∼ 3 times. As depicted in Figure 7, the process begins with MiniGPT-4 generating initial captions from structured prompts. While these captions often correctly identify subjects such as "parking lot" and "vehicle", they sometimes inaccurately describe actions or locations. Annotators then watch the video shot and revise these captions. For instance, errors like "standing in front of", depicted in red, are corrected to "getting close to the car", shown in green. Additionally, annotators enrich captions with key details, such as "a close-up shot of the front of a car." Our single-shot narration caption annotation process follows a similar approach. Differently, we offer ASR text and videos to the annotators and ask them to write down the visually related content and describe the source of the speech. The process during model deployment has been detailed in Sec. 3.2.

## A.2 GPT-4 SUMMARIZATION PROMPT

We utilize GPT-4 to summarize video clips, leveraging our detailed video-shot captions and ASR text. The summarization follows a prompt structure adapted from (Li et al., 2023b), which defines video captions and audio captions for each shot, as depicted in Figure 9. For each video, we organize shot durations, video captions, narration captions, and ASR into a text format (see Figure 8). This arranged content is then fed into GPT-4 for generating the video summary.

## A.3 HUMAN CORRECTION OF VIDEO SUMMARIES

Our detailed shot captions enable GPT-4 to effectively identify and link subjects across shots, without requiring extra re-identification modules. However, according to human evaluation, about 30% of video summaries struggle to connect objects and scenes across shots. Our annotators review these summaries alongside the video clips to correct such errors. Figure 10 illustrates this process. While GPT-4 accurately references the same location, such as "the open field" and "the same open field", it sometimes fails to maintain continuity with elements like "the black car" across scene transitions. Annotators must watch the video and assess the initial summary to make necessary corrections for the final summary. This method ensures the production of high-quality video summaries with efficiency.

## A.4 ANNOTATION OF VIDEO QUESTION-ANSWERING PAIRS

As shown in Figure 13, our annotation of question-answering (QA) pairs utilizes a hybrid manual-automatic approach to ensure both diversity and quality. Initially, we employ GPT-4 to generate

The video has **4** shots. It has **15.8 seconds** in total.
The first action segment starts from **0.0** seconds to **6.5** seconds.
Visual caption of this clip is: **It's a man in a kitchen chopping something with a knife. The man is wearing a tee shirt with the number 19 on it. He seems to hurt himself and appears in pain. The video is in black and white. The yellow words "DON'T RISK IT" appear on the screen. Then the scene changes to a bowl of guacamole with chips on the side. The video also shows the package.**
The audio caption of this clip is: **In the audio, the background voice says don't risk getting injured from cutting up avocados.**

The second action segment starts from **6.5** seconds to **10.5** seconds.
Visual caption of this clip is: **It's a close-up of a person scooping avocado out with a spoon on a wooden cutting board. There is also a white bowl next to it.**
The audio caption of this clip is: **In the audio, the background voice says the product is refrigerated and pre-scooped for the customers.**

The third action segment starts from **10.5** seconds to **13.5** seconds.
Visual caption of this clip is: **It's a woman in a pink shirt with a cast on her arm. She holds her thumb up with the injured arm. The word "SWEET" in purple is typed on the screen.**
The audio caption of this clip is:

The forth action segment starts from **13.5** seconds to **15.8** seconds.
Visual caption of this clip is: **It's a bowl of guacamole with chips on the side. There is a plastic container of guacamole with a purple package next to it.**
The audio caption of this clip is:

The ASR of the video is: **ASR content**

Figure 8: Example of textual content for video in Figure 1. Texts in color are specific for input video and are replaced during our generation.

You are a chatbot that conducts conversations based on video contexts. You mainly answer based on the given caption contexts from different sources, such as visual caption and audio caption. You can also answer the relevant knowledge of the person or object contained in the video. The video has a high-level topic and the video content is supposed to be coherent. The video can have more than one shot and in each shot different action segments and events exist. Visual caption is provided by a validated annotater and it usually describes more details in the video. Audio caption is the caption of audio in the shot, which is relevant to the visual content and it is provided by a validated annotator. Different shots are annotated separately, but the content and topics of different shots are supposed to be coherent. You should refer to the audio caption and extend the content accordingly. You don't create a video plot out of nothing. The video content is:

**{video content}**

Describe the video in detail and coherently, including the actions and scenes. You should not mention the number of scenes, but you ought to describe the content changes and transition. You ought to describe the content in happening and reasoning order. You should not create a video plot out of nothing.

Figure 9: Prompt template for GPT-4 summarization.

candidate question-answer pairs based on our specific instructions, focusing on creating temporal-related, holistic understanding, and audio-related questions. Subsequently, annotators review these pairs while watching the corresponding videos to eliminate simple or incorrect entries. Subsequently, the annotators are asked to categorize the QA pairs into predefined categories, *i.e.*, temporal-related, holistic understanding and audio-related, and discard the data not under these categories.

The prompts used for generating candidate QA pairs are detailed in Figure 11 and Figure 12. For different task-specific questions, we adopt different task instructions as shown in Figure 11. During generation, we adopt the template shown in Figure 12. The boldfaced texts, such as "**{shot_caps}**", "**{video_sum}**", "**{task_inst}**", are replaced with shot captions organized as in Figure 8, video summary and task-specific instructions from Figure 11.

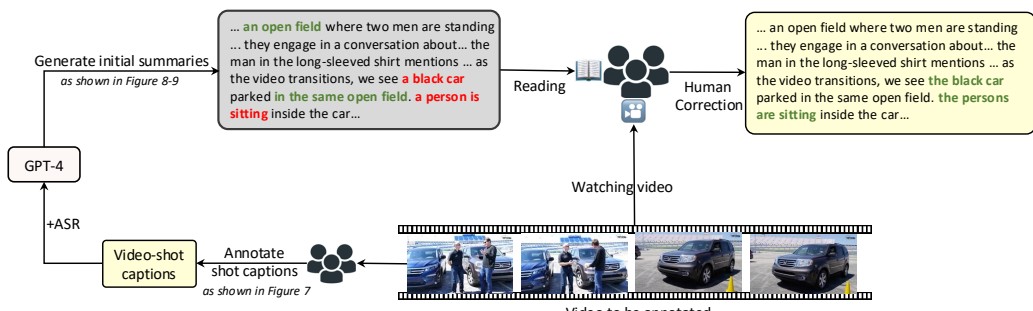

Figure 10: Human correction process of video summaries. Overlapped text is omitted for clarity. Texts in bold green represent correct content, while those in red indicate errors. Please find more explanations in Sec. A.3.

Please ask 5 questions only related to visual content. Please remember to avoid audio content. Please answer the questions according to the Holistic Video Summary, or Video Shot Captions, depending on the question types.

The questions should be related to multiple video shots, rather than a single shot. They can only be answered with multiple video shots and cannot be answered by a single shot. The questions should be logical, reasonable and non-trivial. Please only return json of the questions and answers in the below format.

(a) Temporal-related task instruction

Please ask 5 more new questions only related to visual content. Please remember to avoid audio content. Please answer the questions according to the Holistic Video Summary, or Video Shot Captions, depending on the question types.

The questions should be related to holistic video understanding, such as the topic, the progression and the story line. The questions should be logical, reasonable and non-trivial. The questions should only be answered by viewing and understanding the full video content. Please only return json of the questions and answers in the below format.

(b) Holistic understanding task instruction

Please ask 2 to 5 new questions that can only be answered together with visual and audio contents. Please remember to avoid audio content. Please answer the questions according to the Holistic Video Summary, or Video Shot Captions, depending on the question types.

The questions should be that they can not be answered without audio content. The answer can only be obtained by associating visual content and audio content. Easy questions are strictly prohibited. Remember: Objective questions are always preferred. Please avoid general questions and general answers. General questions are questions that are not specific to the video. General answers mean answers can provided without viewing the video content. An example question is: whom in what appearance speaks what content? This is only for illustration.

(c) Audio-related task instruction

Figure 11: Task instruction for candidate question-answering pairs generation.

# B    PROMPTS USED IN OUR MODELS

In this section, we elaborate on the prompts used for training and testing our models. We detail the prompts for single-shot video captioning and narration captioning in Sec. B.1. For video summarization models like SUM-shot, SUM-holistic, and SUM-text, the prompts are thoroughly explained in Sec. B.2.

## B.1    PROMPT FOR SINGLE-SHOT VIDEO CAPTIONING

During the training of our single-shot captioning models, we select a random text prompt for each video shot, with different model variants utilizing distinct prompts. The prompts for the single-shot video captioning model that incorporates both visual signals and ASR are depicted in Figure 14. In the figure, boldfaced text, such as "{asr}", is replaced with specific video information. The arrangement of visual tokens and text prompts, as presented in Figure 4, is not included here for brevity.

You are a professional video creator. You are required to ask valuable and non-trivial questions according to the given video. Please be notified that the video has multiple shots, containing event progression and storyline. There's also a possibility that the video contains the same objects, subjects and scenes in different video shots.

START VIDEO!

**{shot_caps}**

The video content in the format of Holistic Video Summary:
**{video_sum}**

END VIDEO!

START QUESTIONING!
**{task_inst}**

NOTE:
0. Please ask questions that answers are definite. You can ask questions such as major event, theme or topical category. But you should avoid questions that can be easily answered in logic and content.
1. Please avoid asking repeated questions.
2. Please do not mention the specific video shot index. It is better to describe the starting and ending of a period of content by former and latter actions, events, and objects. Of course, you can use timestamps instead.
3. The questions together should be able to be composited to express the original video content, including event content, event progression, subject appearance and topics.

The questions can definitely be answered by the textual video content. Please generate answers for the questions based on the provided textual video content. The answers should be grounded on the video content, concisely.

Figure 12: Prompt template for candidate question-answering pairs generation.

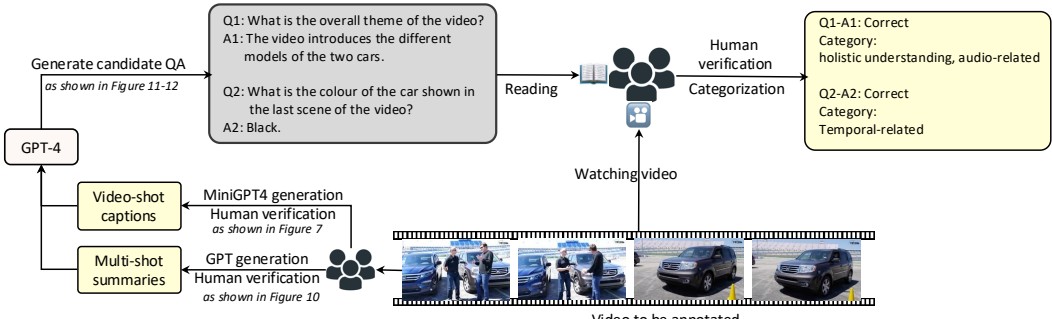

Figure 13: Human annotation process of QA pairs. Please find more explanation in Sec. A.4.

## B.2 PROMPT FOR VIDEO SUMMARIZATION

In Sec. 3.3, we explore different model variants for video summarization, namely MiniGPT4-SUM-shot, MiniGPT4-SUM-holistic, and VideoChat2-SUM-shot. For training these models, we use the same text prompt as in single-shot video captioning, shown in Figure 14. The key distinction between MiniGPT4-SUM-shot and MiniGPT4-SUM-holistic lies in the arrangement of visual tokens: MiniGPT4-SUM-shot incorporates shot-specific information such as shot number or index along with visual tokens from each shot, whereas MiniGPT4-SUM-holistic uniformly samples frames across the video. For VideoChat2-SUM-shot, the input prompt is the same to MiniGPT4-SUM-shot.

## B.3 PROMPT FOR IN-DOMAIN VIDEO QUESTION-ANSWERING

In our paper, we propose a unique question-answering procedure in which we generate video summaries and prompt an LLM to answer the corresponding question. The text prompt used for LLM is shown in Figure 15.

Random one during training:

> 1. The audio transcripts are: **{asr}**. Describe this video in detail.
> 2. In the audio, I hear that: **{asr}**. Take a look at this video clip and describe what you notice.
> 3. Based on the audio, the speech content is: **{asr}**. Please provide a detailed description of the video.
> 4. From the audio, I gather the content that: **{asr}**. Could you describe the contents of this video for me?

Figure 14: Prompt during training for single-shot video captioning.

> I need your help to identify a specific object, place, person and way in the video based on its description. You must answer my question concisely. The answer is definitely contained in the provided video description. Video content is:
>
> **{video summary}**
>
> You should answer the question concisely. Based on the video, please answer **{video question}**

Figure 15: Prompt used for video question answering with summaries.

### B.4 PROMPTS USED FOR ZERO-SHOT QA

In this section, we detail the prompts employed for the zero-shot video QA task, which is discussed in Sec. 3.4. Note that we use the same prompt with Vicuna for text summary-based video question-answering, both zero-shot video QA and in-domain video QA.

#### B.4.1 LLM QA PROMPT

The text prompt is shown in Figure 15, the same to in-domain video QA. It requires video summaries, which are generated from models trained with our Shot2Story data, which fullfils the definition of zero-shot tasks. To better align with the ground truth answers, we prompt the LLM to generate concise answers solely based on the provided video content.

#### B.4.2 EVALUATION PROMPT

In our paper, we follow the same evaluation procedure as outlined in (Maaz et al., 2023), using ChatGPT-3.5 to assess the alignment of the generated answers with the given ground truth.

### B.5 GPTV GENERATION PROMPT

The prompt used for GPTV generation for another 90K videos is shown in Figure 16. We follow a similar prompt structure as Figure 8 to organize the different shots. In addition to the shot structure, we additionally embed speaker information and ASR texts for visual-audio correlation in generated summaries. Using Whisper-X, we extract the speaker diarization and ASR texts, which are organized in the form of "*somebody* speaks *something* during *when* to *when*". The speaker diarization descriptions are then appended to the tail of video content in the prompt, in order to have speaker identification information in the generated summaries. We show two samples in Figure 17, which shows the successful speaker assignment to the visual elements and adequate details. Note this subset of data is not verified by human annotators. We only used it in the video summarization experiments in Section 3.3 of the main paper.

## C ADDITIONAL EXPERIMENT RESULTS

### C.1 COMPARISON OF DIFFERENT MODELS

In this subsection, we present an example of video summarization from our Shot2Story testing split in Figure 18, using MiniGPT4-SUM-holistic, MiniGPT4-SUM-shot, and VideoChat2-SUM-shot. Both MiniGPT4-SUM-shot and VideoChat2-SUM-shot, with their access to shot information including shot count and visual tokens in shots, successfully capture the video's storyline and transitions. For instance, they accurately depict the sequence involving a woman in the kitchen, almonds in an oven,

You are a chatbot that conducts conversations based on video contexts. These are frames of a short video. You mainly answer based on the given frames. You can also answer the relevant knowledge of the person or object contained in the video. The video has a high-level topic and the video content is supposed to be coherent. The video can have more than one shot and in each shot different action segments and events exist. Do not include details that you are not sure of.
Please note that some speakers in the audio appear in the video, of whom the speech content should be described in the shot. Please also note that some speakers may not appear in the video, who may be background voice or camera holders, of whom the speech content should be described as narrator or background voice.

The video has **{num_shots}** shots. Each shots may contain multiple actions, scenes and subjects.
The **{shot_idx}** shot starts from **{start_duration}**s to **{end_duration}**s.
It contains frames **{frames}**
The ASR contained in this shot is: **{shot_ASR)**
*… # more shots in the same structure*

**{speaker_diarization}**

Please create a detailed description that outlines the key actions and components. Please describe the appearance, clothing, and surrounding environment of the characters. Also, please describe the appearance and characteristics of key objects. When multiple people and objects appear, please describe them all and make their descriptions as unique as possible. You should ensure the description is in narrative style and third-person view. You should describe the video coherently

You should describe and include the speech content into each video shot. You should be aware that the speakers in the audio may not appear in the video. If the speakers in the audio does not appear in the video, you should still mention the speech content if it is related to the visual content and topic. The speech content should be properly rephrased from its original ASR texts, if possible. You should ignore the incomplete speech content. Do not mention the words of "ASR". DO NOT include a conclusion of ASR or speakers.
You should not mention duration of videos and video shots. You should not mention the number of scenes, but you ought to describe the content changes and transition. You ought to describe the content in happening and reasoning order. You should not create a video plot out of nothing. Please describe the video coherently. You should not include a separate conclusion paragraph.

Figure 16: Prompt for video summary generation using GPTV.

and the woman speaking to the camera. VideoChat2-SUM-shot, with its advanced vision backbone and video pretraining, captures more nuanced action details, like "using a wooden spatula to roast almonds in an oven". MiniGPT4-SUM-holistic, while effectively identifying major content and events, falls short in accurate scene sequencing due to its lack of shot-structured visual tokens. This leads to errors in narrative order, such as reversing the scenes of "shifts back to the woman in the kitchen" and "in the final scene, the woman is seen using a wooden spatula".

## C.2 QA SUMMARY

In this subsection, we present the results of zero-shot video question-answering using Vicuna v0-13B, based on textual summaries of video samples from MSRVTT-QA (Xu et al., 2017) and ActivityNet-QA (Yu et al., 2019). Despite the limitations of our summarization model, which scores 8.6 in CIDEr on the Shot2Story test split (see Table 3), and the inherent challenges of the videos due to out-of-domain topics or extended durations, the summaries generated from our trained SUM-shot model largely succeed in capturing the key elements of the videos and providing relevant information.

**MSRVTT-QA:** For instance, in Figure 19, `video7089` from MSRVTT-QA portrays a TV show outside the domain of Shot2Story . This genre typically features minimal movement within individual shots, and frequent scene transitions, but a restricted variety of scenes. Yet, our generated summary aptly identifies principal elements such as the judges and contestants, actions like "engaged in a conversation" or "picking up a guitar", and the setting of an American Idol audition. These details equip the summary to competently address questions from MSRVTT-QA. However, some gaps in detail lead to inaccuracies: of the first 10 questions for `video7089`, 5 are incorrectly answered due to missing information (*e.g.*, Q1, Q3), incorrect summary content (*e.g.*, Q4), or misalignment with the ground truth (*e.g.*, Q6, Q10).

**ActivityNet-QA:** In Figure 20, we present the video `v_mZYWfmsYQPA` from ActivityNet-QA. The video's duration is 104 seconds, which is significantly longer than the average duration in our Shot2Story . Our summary effectively identifies important elements such as the main subject's clothing described as "dressed in a black polo shirt with a logo", the actions including "address the

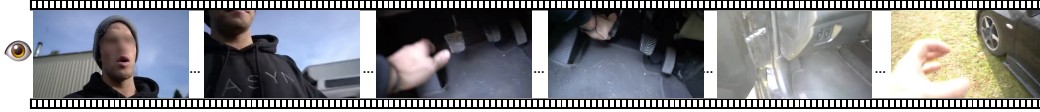

***Speaker diarization***: There is 1 speaker. SPEAKER_00 speaks during 0.01 seconds to 15.3 seconds.

***Human speech***: what exactly is going on. So there's still pressure on the clutch pedal. Like the clutch feels like it works. It's just super, super soft. I know it's not a throw bearing clip issue. Like we all know Evos have their pole style clutch. So there's

***GPTV-generated*** video summary:

In the first shot of the video, a man appears to be outside during daylight, with a clear sky above him. He wears a casual, dark hoodie with the letters "ASYN" on it and a grey beanie. He seems puzzled, his brows furrowed as he questions what exactly is happening, which implies confusion or concern regarding a situation off-camera. Transitioning into the second shot, focus shifts to the inside of a vehicle, where the man's hand is interacting with the clutch pedal, pressing it down and articulating it to test its feel. He expresses that while the clutch pedal still has pressure, it feels unusually soft. The footwell of the vehicle is shown, highlighting the clutch pedal as a key object of interest. We see the hand pressing and wiggling the clutch pedal, conveying an examination of its functionality. In the final shot, the narrative continues with a further detailed explanation about the car's mechanical state. There's an assertion that it is not an issue with the throw bearing clip, a component known to be problematic in certain car models, which are alluded to be "evos" known for their pull-style clutches. The focus remains on the vehicle interior and briefly moves to the exterior, suggesting the man has stepped out, possibly inspecting other parts of the car relevant to the clutch issue. The final frame ends with the man's hand gesturing, emphasizing his point about the car's mechanical specifics, possibly indicating the clutch's pole style.

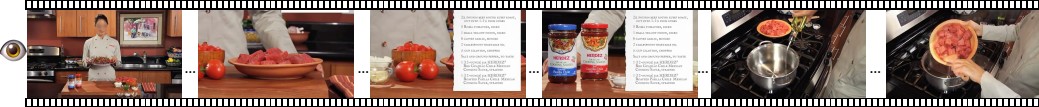

***Speaker diarization***: There is 1 speaker. SPEAKER_00 speaks during 0.8 seconds to 21.9 seconds.

***Human speech***: Here's what you'll need. Beef rump roast, Roma tomatoes, yellow onion, garlic, oil, a little cilantro, some salt and pepper, and last but certainly not least, Herde's Red Guajillo Chile Mexican cooking sauce and roasted pasilla chili sauce. To start, heat oil over medium-high heat. Now take your beef and season

***GPTV-generated*** video summary:

In the first shot of the video, a woman dressed in a professional chef's coat is standing in a kitchen, holding a large dish that appears to be filled with a prepared meal, possibly a salad. She is poised in front of a kitchen counter adorned with various ingredients and cooking utensils. The kitchen features modern appliances and a decorative painting on the wall. As she stands there, an off-screen speaker begins with, "Here's what you'll need," suggesting that the viewer will be instructed on a recipe or cooking method. Transitioning to the second shot, the focus shifts to the ingredients and equipment required for the recipe. Several frames show a close-up of a bowl filled with cut beef rump roast, which is the primary ingredient. The kitchen counter is neatly arranged with Roma tomatoes, a diced yellow onion, minced garlic, and a clear bottle of oil. Fresh cilantro, salt, pepper, and two jars of Herdez brand cooking sauces, one labeled as Guajillo Chile and the other as Roasted Pasilla Chile, are also displayed. A list of ingredients appears on the screen, detailing the quantities and specific items needed, including "2 ½ pounds beef round rump roast, cut into 1-1½ inch cubes" and the other components mentioned. The speaker narrates the list of ingredients, including the beef rump roast, Roma tomatoes, yellow onion, garlic, oil, cilantro, salt and pepper, as well as the specific Herdez Mexican cooking sauces. The video transitions once more into the third shot, where the cooking process begins. A pot is centered on a stove, and the chef pours oil into it. Following this, the chef seasons the beef with salt, preparing it for cooking. The action implies that the beef will soon be browned in the oil. The speaker's instructions -- to heat oil over medium-high heat and to season the beef with salt -- align with the visual actions, demonstrating the initial steps in preparing the dish. The viewer is now both informed of the ingredients and shown the beginnings of the cooking process, paving the way for the continuation of the recipe..

Figure 17: Samples of video summaries generated by GPTV. In both cases, GPTV successfully correlates the audio speaker and the person in the video, *i.e.* "He expresses that" and "an off-screen speaker begins". Moreover, GPTV-generated summaries capture the overall action and event flow with the help of shot structure, and the essential details in the video.

audiance" and "demonstrate the proper grip", and the setting, noted as "a ping pong table in an indoor sports facility". These comprehensive details enable the Vicuna model to correctly answer most questions from ActivityNet-QA. Despite these insights, some omissions and discrepancies in the summary contribute to inaccuracies in answering. Specifically, four out of ten questions are answered incorrectly due to either a lack of specific details or mismatches with the ground truth, as seen in questions Q2, Q5, Q6 and Q9.

**Shot2Story-QA:** In Figure 21, we present a video sample from our Shot2Story-QA dataset. The questions address audio-related (Q1, Q4), holistic understanding (Q2, Q3), and temporal-related (Q4) aspects. With shot structure enabled in VideoChat2-SUM-shot, the generated video summary includes actions and transitions, such as *begins with a close-up shot of a person holding a pedal*, *transitions to a man*, *then shifts to a wider shot*, *returns to a close-up shot of the man*, and *concludes with a wider shot of the man*. However, the generated summary falls short in capturing detailed information about the actions and their orders (e.g., "Taking out a piece of paper" in Q3), and fails to align visual and audio content (e.g., different components in Q1 which have been mentioned in ASR texts). As a result, Vicuna struggles to answer these questions, highlighting the challenges posed by our Shot2Story-QA benchmark. To further improve the model performance, the summarization

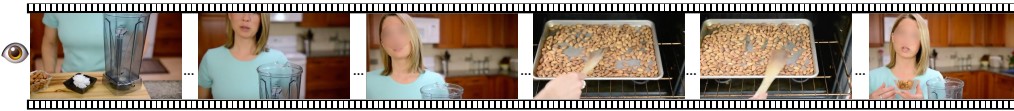

**Human speech**: However, there are a couple of tricks that you need to know in order to be successful. The first is that you must roast your almonds that helps to bring out not only the flavors but the oil as well.

---

*MiniGPT4-SUM-holistic* video summarization:

the video begins with a woman in a well-equipped kitchen, addressing the camera with a smile. she is wearing a green t-shirt and is surrounded by brown wooden cabinets and appliances. as she speaks, she reveals that there are a few tricks that people need to know in order to be successful. the scene transitions to a close-up of a clear, transparent blender. the woman's voice instructs viewers to roast their almonds to bring out not only the flavors but the oil as well. the video then shifts back to the woman in the kitchen, now holding the transparent blender. she continues to speak to the camera, emphasizing the importance of roasting the almonds. in the final scene, the woman is seen using a wooden spatula to scoop up the roasted almonds from a baking tray. she then places the tray in the oven to continue roasting the nuts. throughout the video, the woman's words and actions provide a comprehensive guide on how to successfully roast almonds.

---

*MiniGPT4-SUM-shot* video summarization:

the video begins with a woman in a kitchen setting, wearing a blue top and blonde hair. she is holding a clear plastic container in her hands, which contains almonds. as she speaks to the camera, she mentions a couple of tricks that people need to know in order to be successful. the scene transitions to a close-up of a plate of almonds being cooked in an oven using a shovel. the woman's voice can be heard in the background, explaining that roasting the almonds helps to bring out not only the flavors but the oil as well. in the final scene, the woman continues to hold the clear plastic container filled with almonds. she is seated in a well-equipped kitchen, with a blender placed next to her. as she speaks to the camera, she holds a bowl of almonds in her other hand. the video seems to be a tutorial or a cooking demonstration, with the woman sharing her knowledge and tips on roasting almonds.

---

*VideoChat2-SUM-shot* video summarization:

in the video, a woman with blonde hair, wearing a light blue top, is standing in a kitchen with wooden cabinets in the background. she is holding a glass bowl filled with almonds and a blender. as she talks to the camera, she mentions that there are a couple of tricks that people need to know in order to be successful. the scene then transitions to a close-up shot of a person using a wooden spatula to roast almonds in an oven. the woman's voice can be heard in the background, explaining that roasting the almonds helps to bring out not only the flavors but also the oil. finally, the video returns to the woman in the kitchen, who continues to hold the glass bowl filled with almonds and the blender.

---

*Ground truth* video summary:

the video begins with a woman, dressed in green short sleeves, standing in front of a kitchen console. she is engaged in a conversation, presumably about cooking tips, as she stands next to a bowl of almonds and a transparent mixer. as she speaks, she picks up the bowl of almonds, emphasizing the importance of knowing certain tricks for successful cooking. the scene transitions smoothly to a close-up of an oven pan filled with almonds. a wooden spoon is used to stir the almonds around in the pan. the woman's voice continues in the background, explaining the benefits of roasting almonds. she mentions that this process helps to enhance the flavors of the almonds, suggesting that this is one of the tricks she was referring to earlier. the video concludes with the woman once again standing in front of the kitchen console, continuing her discussion. she holds the bowl of almonds in her hands, possibly preparing to demonstrate the next step in her cooking process. throughout the video, the woman's actions and dialogue suggest that she is sharing her knowledge and tips about cooking, specifically focusing on the use of almonds.

Figure 18: Example for video summarization using MiniGPT4-SUM-holistic, MiniGPT4-SUM-shot and VideoChat2-SUM-shot. All three models effectively grasp the video's main topic and content. Please find explanations in Sec. C.1.

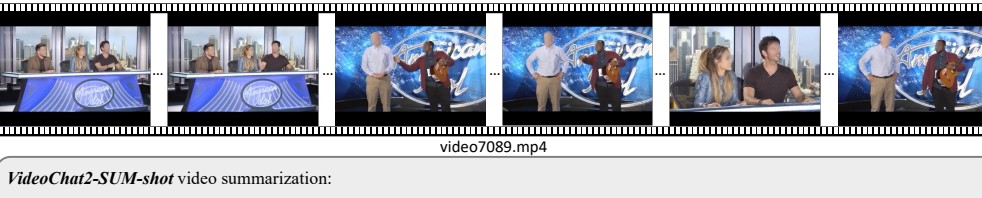

video7089.mp4

**VideoChat2-SUM-shot** video summarization:

in the first shot of the video, three individuals are seated at a desk in what appears to be a television studio with a cityscape view in the background. the man on the left is wearing a dark shirt, the woman on the right is dressed in a light blue blouse, and the man on the right is wearing a black t-shirt. they are engaged in a lively conversation, with the man on the left gesturing with his hands as he speaks. the woman on the right seems to be listening attentively, while the man on the right listens as well. the conversation seems to revolve around a topic related to the city, with the man on the left expressing his enthusiasm for it. transitioning to the second shot, the scene shifts to a stage where two men are standing. the man on the left is dressed in a light blue blazer and a dark shirt, while the man on the right is wearing a dark shirt and khaki pants. the man on the left is holding a ukulele, and they both appear to be in the middle of a performance or presentation. the man on the left is smiling and seems to be enjoying the moment, while the man on the right stands with his hands on his hips, looking at the audience or perhaps the man with the ukulele. in the third and final shot, the video returns to the studio setting with the three individuals from the first shot. the man on the left is still wearing his dark shirt, the woman on the right is in her light blue blouse, and the man on the right is in his black t-shirt. the man on the left is gesturing with his hands again as he speaks, while the woman on the right and the man on the right listen attentively. the woman on the right seems to be responding to something the man on the left is saying, and the man on the right listen intently.

**Questions**:

Q1: who speaks to tv show judges?
Q2: who do three judges talk to?
Q3: what is a black male laughing on?
Q4: what talks to a performer?
Q5: what did the man audition for?
Q6: who does a panel talk to?
Q7: what do judges talk on?
Q8: what shows judges of a singing competition?
Q9: how many contestants are standing before the judges at american idol?
Q10: what is going on?

**Answers**:

GT1: person
GT2: contestant
GT3: idol
GT4: panel
GT5: idol
GT6: performer
GT7: idol
GT8: video
GT9: two
GT10: reality

**Vicuna answers using summaries & GPT evaluation**:

A1: Based on the video description, it is not clear who is speaking to TV show judges. ❌
A2: The three judges in the video are the man in the black shirt and khaki pants, who claps his hands and smiles, and the woman in the blue and silver-colored shirt, who looks at the camera. ✅
A3: I'm sorry, but I cannot identify a specific person laughing in the video, as there is no mention of a person laughing in the description. The video appears to be focused on the interactions and performances of the individuals in front of the American Idol backdrop. ❌
A4: The man in the white shirt and black pants is a performer. ❌
A5: The man in the white shirt and black pants auditioned for American Idol. ✅
A6: Based on the video, it appears that a panel talk is happening between the man in the black shirt and the woman in the blue shirt, who are sitting behind a table with a city skyline in the background. The man occasionally claps his hands, and the scene transitions to two men standing next to a large screen with \"American Idol\" written on it. One man, dressed in a white shirt and black pants, picks up a guitar and starts playing, while the other man, wearing a black shirt and khaki pants, claps his hands and smiles. In the final scene, the man in the white shirt and black pants continues to play the guitar, while the woman in the blue and silver-colored shirt looks at the camera. The man in the black shirt and khaki pants claps his hands and smiles. ❌
A7: Judges talk about performances in front of the American Idol backdrop. ✅
A8: The video shows judges of a singing competition. ✅
A9: Two contestants are standing before the judges at American Idol. ✅
A10: It appears that the video is a behind-the-scenes look at a recording session for a musical performance, possibly for a talent competition such as American Idol. The individuals shown in the video are a group of musicians who are preparing for a performance in front of a panel of judges. The man in the black shirt and khaki pants is likely a music producer or judge, while the man in the white shirt and black pants is likely a contestant on the show. The woman in the blue and silver-colored shirt is likely another member of the group or a crew member. ❌

Figure 19: Example for video question-answering using summaries - MSRVTT. We show the zero-shot video summary using our trained VideoChat2-SUM-shot model. The summary can capture most of the video, such as the subject, the action and the transitions. We also show the answers from Vicuna to the first 10 questions and the evaluation results from GPT. Please find more explanations in Sec. C.2.

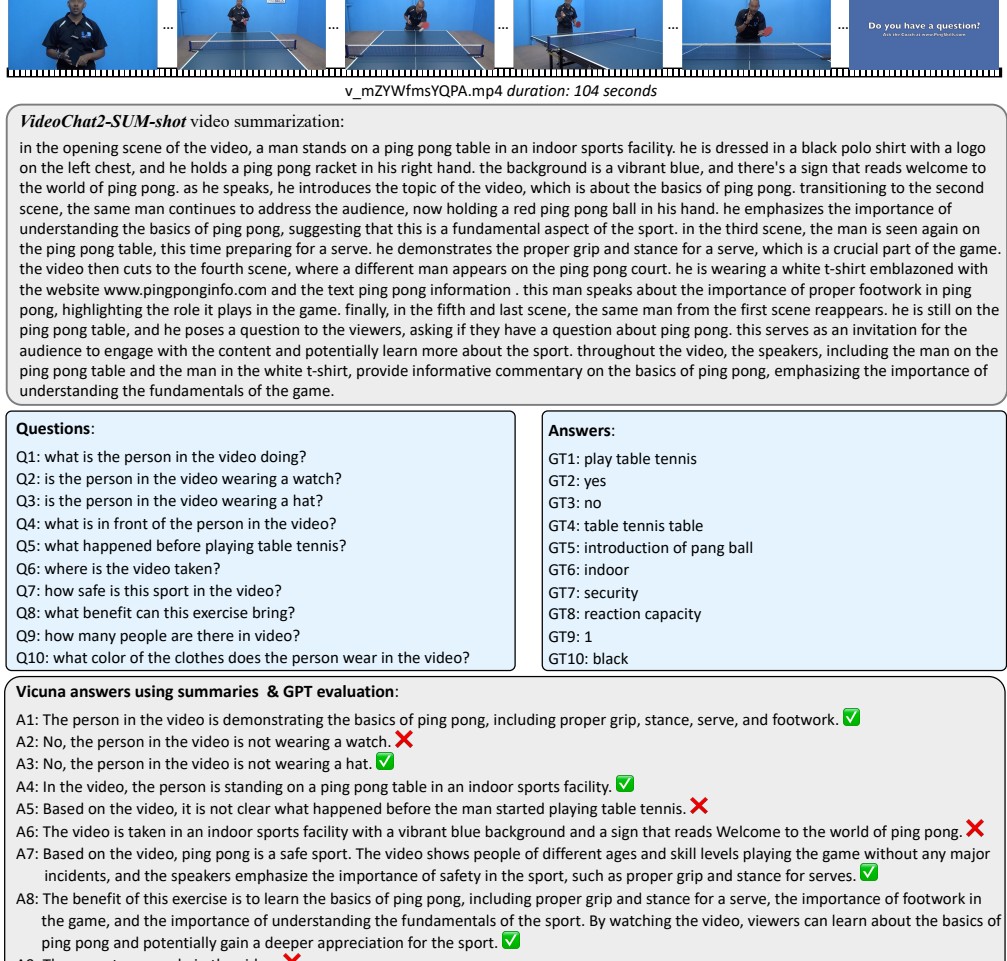

Figure 20: Example for video question-answering using summaries - ActivityNet. We show the zero-shot video summary using our trained VideoChat2-SUM-shot model. The summary can capture most of the video, such as the subject, the action and the transitions. We also show the answers from Vicuna and the evaluation results from GPT. Please find more explanations in Sec. C.2.

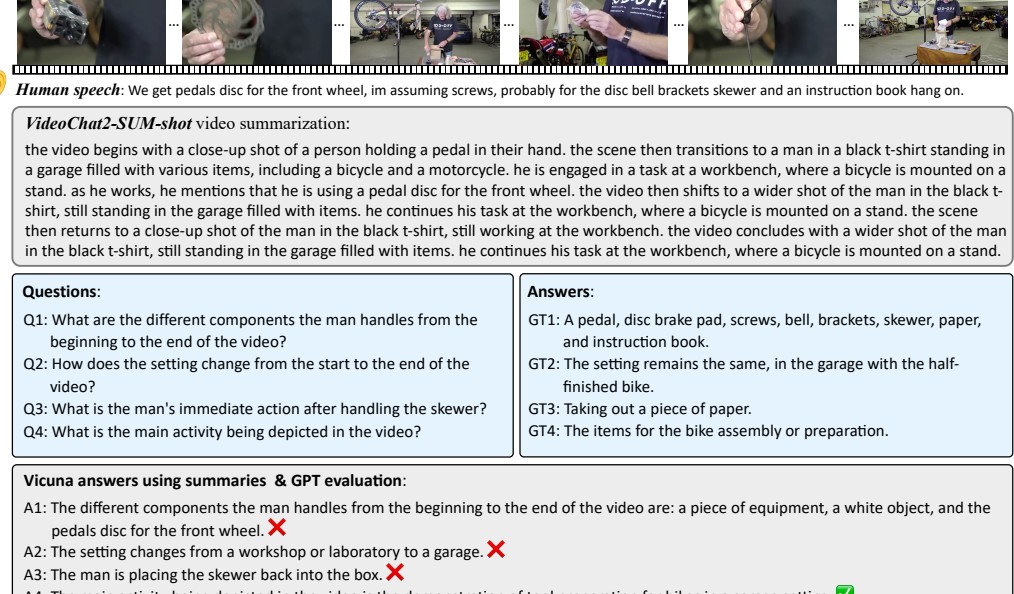

*Human speech*: We get pedals disc for the front wheel, im assuming screws, probably for the disc bell brackets skewer and an instruction book hang on.

*VideoChat2-SUM-shot* video summarization:

the video begins with a close-up shot of a person holding a pedal in their hand. the scene then transitions to a man in a black t-shirt standing in a garage filled with various items, including a bicycle and a motorcycle. he is engaged in a task at a workbench, where a bicycle is mounted on a stand. as he works, he mentions that he is using a pedal disc for the front wheel. the video then shifts to a wider shot of the man in the black t-shirt, still standing in the garage filled with items. he continues his task at the workbench, where a bicycle is mounted on a stand. the scene then returns to a close-up shot of the man in the black t-shirt, still working at the workbench. the video concludes with a wider shot of the man in the black t-shirt, still standing in the garage filled with items. he continues his task at the workbench, where a bicycle is mounted on a stand.

**Questions**:

Q1: What are the different components the man handles from the beginning to the end of the video?

Q2: How does the setting change from the start to the end of the video?

Q3: What is the man's immediate action after handling the skewer?

Q4: What is the main activity being depicted in the video?

**Answers**:

GT1: A pedal, disc brake pad, screws, bell, brackets, skewer, paper, and instruction book.

GT2: The setting remains the same, in the garage with the half-finished bike.

GT3: Taking out a piece of paper.

GT4: The items for the bike assembly or preparation.

**Vicuna answers using summaries & GPT evaluation**:

A1: The different components the man handles from the beginning to the end of the video are: a piece of equipment, a white object, and the pedals disc for the front wheel. ❌

A2: The setting changes from a workshop or laboratory to a garage. ❌

A3: The man is placing the skewer back into the box. ❌

A4: The main activity being depicted in the video is the demonstration of tool preparation for bikes in a garage setting. ✅

Figure 21: Example for video question-answering using summaries - Shot2Story-QA. We show the in-domain video summary using our trained VideoChat2-SUM-shot model. The summary can capture the shot transitions and major actions. However, as indicated by the QA results, the detailed actions and their order fail to match with the groundtruth, such as whether the immediate action after handing the skewer. Additionally, it fails to provide adequate information regarding the consistent scene between shot transitions, such as "at a workbench" and "in the garage". Please find explanations in Sec. C.2.

model could be enhanced with detailed information with denser frames, fine-grained matching of audio and visual cues, and tracking of the same objects across the video. We leave it as future work.

## D  ADDITIONAL IMPLEMENTATION DETAILS

**Video-Shot Captioning.** For each video shot, we uniformly sample 4 frames. For testing, a consistent text prompt is used, *i.e.*, "The audio transcripts are asr. Describe this video in detail.". The maximum number of new tokens generated by the LLM is capped at 180 for both training and inference.

**MiniGPT4-SUM-shot, MiniGPT4-Holistic and VideoChat2-SUM-shot.** In two SUM-shot models, 4 frames per video shot are sampled uniformly. In MiniGPT4-SUM-holistic, 16 frames per video clip are sampled. The rationale behind sampling 16 frames in a holistic approach is based on our dataset's average of 4 shots per video, aligning with the SUM-shot approach of 4 frames per shot. For both training and inference, the LLM's maximum new token count is set at 600. A consistent text prompt, *i.e.*, "The audio transcripts are {asr}. Describe this video in detail.", is used during inference.

For video-shot captioning and video summarization tasks, both models are trained on $8 \times 2$ A100-80G GPUs using Pytorch. The captioning model is trained for 10 epochs, with the best-performing checkpoint on the validation set used for test performance reporting. To prevent overfitting, text prompts are randomly sampled for each sample, as detailed in Sec. B.

**VAST.** We tune the model following the official instructions by inputting our video frames, audios and ASR texts. The optimizing target is the concatenation of our single-shot visual caption and single-shot narration caption. During inference, we input the corresponding modalities, in case of two different model versions we trained (visual + audio, or visual + audio + ASR texts), into the model and use the directly generated captions for evaluation. The model is trained on $8 \times 4$ A100-80G GPUs for 3 epochs, maintaining other hyperparameters at their default values from the original configuration.

**Video-ChatGPT.** Consistent with SUM-holistic, we uniformly sample 16 frames for both training and inference. Our prompt setup excludes ASR for video summarization. The training is conducted over 3 epochs with a learning rate of 2e-6, and we retain other hyperparameters at their default settings as specified in the original repository.

**Video Question-Answering with Summary.** For the MSRVTT-QA (Xu et al., 2017) and ActivityNet-QA (Yu et al., 2019) datasets, we generate video summaries using our trained SUM-shot model, employing only visual tokens during inference. Upon generating these summaries, we integrate them with individual questions from their corresponding videos into the prompt format displayed in Figure 15. This integrated content is then processed through Vicuna (Chiang et al., 2023) to obtain answers. The evaluation of these results is carried out following the methodology outlined in (Maaz et al., 2023).

## E  BROADER IMPACT

**Data Limitations and Ethical Considerations.** We provide cropped multi-shot videos instead of the original videos. Users can also turn to download these from original sources. Given HD-VILA-100M (Xue et al., 2022)'s long-standing public availability, we assess a low risk of the currently available videos being removed in the near future. Additionally, our meticulous manual annotation process is designed to avoid any ethical or legal violations. Specifically, our videos don't have personally identifiable information or offensive content, which is ensured by the manual annotation process. The authors will take the responsibility of long-term maintenance.

**Human Rights in Annotation Process.** We have conscientiously structured the annotation process to ensure fair workloads and equitable compensation for annotators, upholding human rights standards.

**Scope of Conclusions.** It is important to recognize that experiments and data, including ours, might only represent a subset of universal realities. Nevertheless, given the wide range of categories covered in our videos, we believe our conclusions offer a robust understanding applicable to various multi-shot

video scenarios and durations. These findings, while specific to our dataset, provide significant insight into the broader field of video analysis.

**Usage of Language Models.** Our use of the LLaMA model (Touvron et al., 2023) from Meta is authorized for research purposes. Those intending to use our model post-release should ensure that they have the necessary permissions and adhere to usage restrictions. We express deep respect for the work of developers and contributors, recognizing their integral role in advancing language modelling and multi-modal learning.

**Future Research and Development.** We release both our code and dataset. This is intended to encourage further research and enable others to build upon our work. Although our current experiments require up to $8 \times 2$ A100-80G GPUs, we are aware this may be a limitation. Consequently, we plan to focus future efforts on adapting these experiments to be compatible with a single node of 8 A100 GPUs. It's important to note that fitting the experiments within an 8 GPU framework is not the primary focus of this paper, but we consider it a crucial step towards making our research more accessible and inclusive for a wider array of research groups.