# OpenReview forum: "Shot2Story: A New Benchmark for Comprehensive Understanding of Multi-shot Videos"
_ICLR.cc/2025/Conference — ICLR 2025 Poster_

### Official Review · Reviewer_YdUF · 2024-11-02

**Soundness:** 3
**Presentation:** 3
**Contribution:** 3
**Rating:** 6
**Confidence:** 2

**Summary:**

The paper presents a new benchmark for understanding multi-shot videos, named Shot2Story. The benchmark provides a dataset of 42,958 short videos, each consisting of an average of 4.4 shots, with detailed textual descriptions, video summaries, and question-answering pairs for multi-modal understanding. It aims to enhance the understanding of videos by providing shot-level captions, human narrations, and generated summaries. Several distinct tasks are defined using this dataset: single-shot video captioning, multi-shot video summarization, and multi-shot video question answering. The paper also presents baseline models and demonstrates the challenges of long, comprehensive video summaries.

**Strengths:**

1. The paper is well-written, providing clear explanations of the benchmark, tasks, and dataset construction, making complex ideas accessible.
2. Shot2Story sets a new standard for multi-shot video understanding, with detailed shot-level captions, audio narrations, and summaries that advance research in video analysis and multi-modal understanding.
3. The benchmark's unique focus on multi-shot videos with rich shot-level annotations differentiates it from previous benchmarks, contributing to a better understanding of event transitions in multi-shot videos.
4. The authors conduct thorough experiments using baseline models like MiniGPT-4 and VideoChat2, providing valuable insights into current models' strengths and limitations, and establishing a solid foundation for benchmarking future models.

**Weaknesses:**

1. There is a format issue in Table 1 that affects its readability. Consistent formatting is essential for clarity and for the accurate presentation of comparative data.
2. The motivation for annotating audio text descriptions is not clearly articulated. Understanding audio alongside video is crucial for comprehensive multimedia understanding, especially in tasks involving multi-modal large language models (MLLMs). Merely using textual descriptions without integrating auditory context reduces the potential for deep reasoning capabilities across modalities, which limits the advancement of MLLM applications.
3. The experimental section lacks baselines from well-regarded models such as the Qwen-VL series. Including such comparisons would provide a stronger benchmark and enable a more effective evaluation of the proposed model’s performance.
4. The paper does not propose any self-developed model tailored specifically for the Shot2Story benchmark, which would demonstrate the specific advantages and limitations of the benchmark through a purpose-built approach.

**Questions:**

1. Is it an optimal approach to use large language models (LLMs) to annotate datasets and then use those annotations for reasoning tasks by the LLMs themselves? This self-referential process might introduce biases or limitations in understanding. Is there a better approach to improving the depth of reasoning and multi-modal alignment, perhaps involving a more collaborative method of training distinct specialized models (e.g., one focused on annotation and another on reasoning)?

---

> ### Author Response · Authors · 2024-11-28
> **Motivation and optimality**
>
> Thanks for your valuable comments. We respond to your concerns below,
>
> 1. **Motivation of annotating audio text description:** We appreciate your insight regarding the integration of audio modalities. Currently, our approach correlates speech semantics with visual content, serving as a foundational step towards multi-modal alignment. While we recognize the importance of incorporating auditory context for comprehensive multimedia understanding, this integration remains part of our future work. Our focus on visual-oriented alignment in the present study provides a structured basis upon which we aim to build more complex multi-modal reasoning capabilities.
>
> 2. **Absence of a Self-Developed Model:** Actually, we introduce the SUM-Shot models, explicitly designed for the Shot2Story benchmark. Unlike traditional holistic models that treat videos uniformly, SUM-Shot leverages structured prompting and a multi-shot framework to emphasize temporal sequences and contextual relationships between shots. This specialization allows the model to maintain narrative coherence and accurately track multiple events and actions across different shots. Our experimental results in Table 3 show that SUM-Shot significantly outperforms standard models, effectively handling the complexities inherent in multi-shot video narratives and showcasing the specific advantages of our benchmark.
>
> 3. **Optimality of using LLMs for annotation and reasoning tasks:** Thank you for addressing this important consideration. Our primary motivation is to demonstrate that the detailed summaries generated through our approach are versatile and effective across multiple downstream tasks. This effectiveness inherently signifies that our annotations are of high quality, providing a robust foundation for various applications in multi-modal understanding. To achieve this, we employed an off-the-shelf LLM for reasoning tasks.
>
>     We acknowledge that a more collaborative approach—such as utilizing distinct specialized models for annotation and reasoning—could mitigate biases and further enhance multi-modal alignment, developing and integrating such systems falls outside the scope of our current work. Nevertheless, we recognize the value of this approach and consider it a promising direction for future research.

---

> > ### Comment · Reviewer_YdUF · 2024-12-02
> >
> > The author addressed my concerns, so I will keep the positive score.

---

### Official Review · Reviewer_n4p7 · 2024-11-02

**Soundness:** 3
**Presentation:** 3
**Contribution:** 3
**Rating:** 6
**Confidence:** 4

**Summary:**

For multi-shot videos, current video benchmarks tend to annotate captions in a coarse-grained manner, either by providing a holistic caption for the entire video or by allowing annotators to subjectively define the boundaries of each event. But multi-shot videos often include rich narrations that correspond to distinct events occurring within the video. Given this context, this work introduces a new benchmark, *Shot2Story*, aimed at enhancing audio-visual understanding of multi-shot videos through the detailed textual annotation of each individual shot. The authors conduct extensive experiments from three perspectives: single-shot video captioning, multi-shot video summarization, and video question-answering with video summaries. These experiments demonstrate significant limitations in current models’ abilities to understand multi-shot content. Additionally, the authors design two baseline models based on MiniGPT-4 and Video-Chat2. Experimental results reveal that the multi-shot instruction data constructed by the authors substantially improves the performance of multi-shot video understanding.

**Strengths:**

This paper addressed here underscores a critical issue in current Video LLM work, where shot-level temporal understanding in video instruction data has largely been neglected. Most existing video instruction datasets lack the fine-grained, shot-level annotations necessary to capture nuanced transitions between events within videos.

To address and evaluate this issue, this work introduces a challenging multi-shot video understanding instruction dataset and benchmark. The dataset construction process involves substantial human oversight, which significantly reduces the likelihood of inaccuracies and biases often introduced by external models (e.g., GPT).

In the experiments, the authors analyze and compare several existing Video LLM approaches, highlighting the benchmark’s applicability and challenges. Additionally, they propose two suitalbe baseline models ,which achieve considerable improvements in the understanding of multi-shot video content.

**Weaknesses:**

The methods compared across the three tasks—video shot captioning, video summarization, and multi-shot video question answering—lack consistency, making it difficult to observe a clear, uniform improvement in performance across these tasks with the proposed dataset. To improve consistency, I suggest that the authors consider using the same set of baseline models across all tasks.

The constructed multi-shot dataset appears to resemble a series of single-shot descriptions loosely connected by simple transition words, such as "begin with," "then," and "next." The annotation process seems more akin to first splitting multi-shot videos into individual shots, generating isolated descriptions for each, and then using an external tool (e.g., GPT-4) to concatenate these descriptions. This method does not accurately represent the interconnections between shots, which may pose challenges in teaching large language models (LLMs) multi-shot temporal reasoning skills using the constructed instructional data. Although the actual content contained in different shots is inconsistent, there is an obvious time sequence and event relationship between them. I suggest that the authors improve the annotation process by having annotators explicitly describe relationships or continuity between shots, rather than relying on simple transition words. This approach would better capture the temporal flow and coherence between shots, which is essential for developing multi-shot temporal reasoning capability in LLMs.


Moreover, it raises the question of whether multi-shot understanding is at odds with simple video understanding and long-video comprehension. I noticed that the average length of videos in the dataset is only 17.1 seconds, and shots with minimal variation between them were deliberately excluded during construction. Although the videos are relatively short, they contain numerous individual shots with significant differences between them. This could lead models to develop a bias toward segmenting any input video into multiple shots for interpretation, which may pose challenges for understanding both simpler and longer videos. I recommend that the authors conduct additional experiments to evaluate whether models trained on this dataset struggle with simple single-shot videos or longer videos, as this could help clarify any potential biases introduced by the dataset construction.

**Questions:**

See weaknesses.

---

> ### Author Response · Authors · 2024-12-03
> **Response**
>
> Thanks for your constructive feedback. We respond to your concerns as below
>
> 1. **Baseline models across all tasks**: We appreciate your observation regarding the consistency of baseline models across the three tasks. To clarify, we employed the same set of baseline models—MiniGPT-4 and Video-Chat2—across all tasks, including single-shot video captioning, video summarization, zero-shot question answering, and multi-shot video question answering, as detailed in Table 2 to Table 5. This consistent application ensures comparability and demonstrates the robustness of our Shot2Story dataset. The experimental progression follows a hierarchical training approach: video summarization models are pretrained on single-shot captions (Table 2), and subsequently, question-answering is performed with the video summaries from models in Table 3. This inherited and progressive methodology underscores the utility of our multi-shot annotation data in enhancing large video-language model training and highlights the high quality of our annotations. The consistent use of baseline models across all tasks facilitates clear and uniform observation of performance improvements, validating the effectiveness of our dataset in advancing multi-shot video understanding.
> 2. **Annotation of transition between shots**: We acknowledge the importance of capturing meaningful transitions and relationships between shots to enhance multi-shot temporal reasoning in large language models. Our dataset annotations have actually annotated detailed descriptions of shot transitions and progressions. These include explicit annotations of camera movements such as "camera zooms in/out," "camera pans," and "video cuts/shifts to," as well as progressive narrative phrases like "as the video progresses," "the video continues," and concluding statements like "video wraps with." These annotations go beyond simple transition words by providing specific information about the nature of the transitions and the continuity between shots, thereby ensuring a coherent temporal flow and interconnectivity. This structured approach effectively captures the temporal dynamics essential for developing robust multi-shot temporal reasoning capabilities in LLMs.
> 3. **Understanding simpler and longer videos**: We have conducted experiments using the ActivityNet-QA dataset, which comprises much longer videos with an average duration of approximately 10 minutes and contains continuous shots with minimal variations. As shown in Table 4, our models, trained exclusively on the Shot2Story multi-shot videos, achieved superior performance in generating video summaries compared to existing instruction-tuned models. This indicates that our models can effectively handle both simpler and longer videos without being adversely affected by potential biases introduced during training. Our approach to processing long videos involves utilizing the same pipeline as for multi-shot videos, to approximate detect shot boundaries (due to motions and variations), resulting in only a few shots (considering the much longer durations, compared to our multi-shot videos in averaged 17 seconds). These findings demonstrate that training on the Shot2Story dataset enhances the model's capacity to understand and process videos of varying lengths and complexities, thereby mitigating concerns about potential biases.

---

### Official Review · Reviewer_G1Gp · 2024-11-04

**Soundness:** 3
**Presentation:** 3
**Contribution:** 3
**Rating:** 6
**Confidence:** 4

**Summary:**

- This paper presents Shot2Story, a large-scale benchmark for comprehensive multi-shot video understanding.

- The dataset contains detailed shot-level captions for both visual signals and human narrations, and comprehensive video summaries based on shot-level captions.

- The authors further design a challenging video question-answering benchmark for multi-shot video understanding.

**Strengths:**

- The paper presents a new multi-shot video understanding benchmark (Shot2Story), with detailed shot-level captions, comprehensive video summaries and question-answering pairs. This will benefit the community, as a benchmark for multi-modal, multi-shot video understanding benchmark.

- The paper has made significant efforts for curating the datasets, to maintain high quality standard, getting the data from 2.1M video clips to 42,958 video clips. The descriptions of the filtering stages are clear and reasonable.

- For baseline establishment, the authors have extensively evaluated a series of open-source models.

**Weaknesses:**

- My main concern is on its scientific novelty, as the paper does not include new designs on tackling the multi-shot video evaluation problem, only providing a benchmark, it seems to be more suitable for a benchmark track.

- For baseline evaluation, the focus has been mainly on open-source models, what about models, for example, gpt4v, claude3.5 ?

- In Table 1, \hline is incorrect ?

**Questions:**

I think the main concern is on its contribution of methodology, as it doesn't involve any architecture/method design for tackling the multi-shot video understanding problem.

---

> ### Author Response · Authors · 2024-11-25
> **Novelties**
>
> Thanks for the comments. Please check our replies to your concerns below:
>
> 1. **Scientific novelty and contribution:** We would like to clarify that ICLR does consider benchmark and dataset papers for publication, as highlighted in the Call for Papers for ICLR 2025. Benchmarks play a pivotal role in advancing research domains by providing standardized datasets that facilitate consistent evaluation and comparison of different models. Our contribution lies in the creation of Shot2Story, a large-scale, meticulously curated benchmark that addresses the complexities of multi-shot video understanding through detailed shot-level captions, comprehensive video summaries, and a challenging question-answering (QA) component.
>
> 2. **Novelty of Shot2Story:** Shot2Story distinguishes itself by integrating multiple modalities—combining audio narratives with visual content—to provide a richer and more nuanced understanding of videos beyond mere textual descriptions. This multimodal approach allows models to leverage both auditory and visual information, thereby facilitating more accurate and contextually relevant video summaries and QA responses. Additionally, Shot2Story introduces tasks that assess models on both captioning and QA, highlighting their ability to comprehend and generate descriptions as well as to reason about the content in a question-answering framework. Our benchmark also showcases superior zero-shot video question-answering performance, underscoring its potential to drive advancements in video understanding.
>
> 3. **Architecture design for multi-shot video understanding:** Beyond providing a comprehensive benchmark, we introduce the SUM-Shot models specifically designed for multi-shot video understanding. Unlike traditional holistic models that process videos uniformly, SUM-Shot utilizes structured prompting and a multi-shot framework to emphasize the temporal sequence and contextual relationships between shots. This design enables the model to maintain narrative coherence and accurately track multiple events and actions across different shots. Our experiments demonstrate that SUM-Shot models significantly outperform standard models, effectively handling the complexities of multi-shot video narratives.

---

> ### Author Response · Authors · 2024-12-03
> **Follow-up**
>
> Dear Reviewer G1Gp,
>
> We deeply appreciate the constructive comments and the comprehensive review you’ve provided. As the rebuttal period concludes soon, we would like to confirm whether our responses have addressed your concerns. If any further clarification is necessary, we would be happy to discuss additional details with you.
>
> Kind Regards!

---

### Official Review · Reviewer_5bm5 · 2024-11-06

**Soundness:** 3
**Presentation:** 3
**Contribution:** 2
**Rating:** 5
**Confidence:** 4

**Summary:**

The paper introduces Shot2Story, a benchmark aimed at improving multi-shot video understanding through a rich set of annotations, including shot-level captions, detailed video summaries, and question-answer pairs. The benchmark focuses on tasks such as single-shot captioning, multi-shot summarization, and multi-shot question-answering, each designed to leverage both visual and auditory cues in video data. The authors provide a comprehensive dataset of nearly 43,000 videos, annotated with visual and audio information at each shot level, as well as summaries and QA pairs to facilitate high-level semantic understanding of video content.

**Strengths:**

Shot2Story addresses the limitations of existing benchmarks by providing fine-grained shot-level annotations, narration captions, and comprehensive video summaries. This detailed annotation setup allows for a more nuanced understanding of multi-shot videos.

The dataset's scale and quality are commendable, with 42,958 videos annotated both visually and auditorily. The authors employ advanced techniques, including GPT-4 for generating initial summaries, followed by human verification, ensuring data reliability.

The authors present solid baseline models across tasks, demonstrating the benchmark's challenges and providing insights into areas where current models struggle, particularly in complex video summarization.

**Weaknesses:**

The reliance on automated generation may introduce inaccuracies or biases, despite human verification. Clarifying any common pitfalls in the generated summaries would strengthen the paper.

The baseline models appear to struggle with integrating visual and auditory information effectively, as noted in the single-shot captioning task. It may be beneficial to explore more advanced audio-visual fusion models as baselines.

A deeper analysis comparing Shot2Story with related benchmarks would contextualize its contributions, especially in terms of unique challenges posed by shot transitions and multi-event progression.

**Questions:**

The paper mentions that models sometimes produce hallucinated details in the video summaries. Did you explore any methods to mitigate these issues during model training or fine-tuning? Additionally, were there specific patterns observed in these hallucinations (e.g., adding non-existent objects, exaggerating actions)?

The distinction between single-shot captioning and multi-shot summarization is well-explained. However, were there cases where summarizing at the shot level led to information redundancy or fragmentation when compiling multi-shot summaries? If so, what strategies did you use to address this issue?

---

> ### Author Response · Authors · 2024-11-25
>
> Thank you for your comments. We reply to your concerns as following,
>
> 1. **Common pitfalls in the generated summaries**: Our dataset predominantly features human-centric activities due to our video filtering process, which selects videos rich in visually related audio information, and the sourcing from HDVILA, which primarily curates content related to human activities. Consequently, our annotations tend to highlight salient events and large-scale objects essential to the video’s storyline, mentioning smaller objects only when they directly contribute to the narrative. Common pitfalls in the generated summaries include the omission of minor yet contextually important details and a bias towards emphasizing more prominent actions or objects, potentially overlooking less conspicuous elements.
>
> 2. **Challenges posted by shot transitions and multi-event progression**: Shot2Story introduces unique and complex challenges that stem from its emphasis on shot transitions and multi-event progression, setting it apart from existing benchmarks like MSRVTT and ActivityNet.
>   - Taking question-answering task as example, unlike MSRVTT-QA, which poses questions such as "Who do three judges talk to?" (Figure 19) , or ActivityNet-QA's general inquiries like "What is the person in the video doing?" (Figure 20), that primarily assess the model's ability to understand interactions at one timepoint, Shot2Story-QA delves deeper into the temporal and causal dynamics of multi-shot videos, by posing "Temporal-related" and "Multi-shot Holistic Understanding" questions (two of our QA question types).
>   - For instance, a temporal-related question like "What is the man's immediate action after handling the skewer?" (Figure 21) requires the model to not only identify specific actions but also comprehend the sequence and progression of events within the video. This necessitates a sophisticated understanding of narrative flow and the ability to link consecutive actions meaningfully.
>   - Moreover, multi-event progression within a single video requires models to accurately capture and represent multiple concurrent events, ensuring that each event is understood in its proper temporal context. Shot2Story-QA's multi-shot holistic understanding question "How does the setting change from the start to the end of the video?" requires the model to understand the multi-shot progression and the event within each shot.
>
> Shot2Story-QA provides a more rigorous and nuanced evaluation framework that pushes the boundaries of current video understanding and summarization models beyond the capabilities assessed by benchmarks like MSRVTT-QA and ActivityNet-QA.
>
> 3. **Information redundancy when compiling multi-shot summaries**: Regarding potential information redundancy when compiling multi-shot summaries, we define redundancy based on the temporal and contextual relevance of shots. Specifically, if two shots with similar content are not consecutive, their inclusion in the summary is not deemed redundant but rather indicative of a coherent event progression. To prevent immediate repetition, our data filtering process ensures that consecutive shots do not contain similar content, as detailed in Lines 129-130 of the manuscript. By maintaining diversity between consecutive shots and preserving the narrative flow across non-consecutive similar shots, we minimize information fragmentation and ensure comprehensive coverage of the video’s storyline. This approach allows us to capture essential elements without unnecessary repetition, thereby enhancing the quality and cohesiveness of multi-shot summaries.
>
> 4. **Video hallucination**: Thank you for highlighting the issue of hallucinated details in video summaries. Although mitigating hallucinations was not the primary focus of our study, we acknowledge its importance and have conducted an analysis of the hallucination patterns observed in our models. We identified two main types of hallucinations: (1) Multimodal Alignment Errors, where the model incorrectly associates visual elements with textual descriptions, such as inaccurately highlighting the "black" color in Figure 5, and (2) Hallucinations with Repeated Details, where similar visual contexts in different shots lead to redundant or incorrect descriptions. As shown in Figure 21, "woman holding a bowl with almonds" only appears in the last shot, while the model erroneously describes the woman as "holding something with almonds" in both the initial and ending shots of Figure 21 SUM-shot models.  Moving forward, we plan to explore advanced multimodal alignment techniques in future research.

---

> ### Author Response · Authors · 2024-12-03
> **Follow-up**
>
> Dear Reviewer 5bm5,
>
> We sincerely appreciate the time and effort you've dedicated to reviewing our paper, as well as your insightful and constructive feedback. As the rebuttal period is drawing to a close, we kindly ask if our responses have sufficiently addressed your concerns. Please feel free to reach out if further discussion or clarification is needed; we are more than willing to engage with you.
>
> Kind Regards!

---

### Meta-Review · Area_Chair_kpCc · 2024-12-21

**Metareview:**

This paper introduces a multi-shot video understanding benchmark, providing captions for both visual signals and human narration. Based on this new benchmark, several tasks are evaluated to establish a foundation for future research.
Initially, the submission received mixed feedback. While reviewers acknowledged the strengths of the proposed benchmark, two reviewers leaned toward rejection. The first reviewer (G1Gp) raised the following concerns: (1) the submission might be more appropriate for the dataset track due to a lack of technical novelty, and (2) it did not include comparisons with non-open-source models such as GPT-4V and Claude 3.5. In the rebuttal, the authors clarified that the submission's primary area is indeed aligned with ICLR’s Datasets and Benchmarks track. This clarification prompted the reviewer to raise their rating to a 6.

The second opposing reviewer (5bm5) did not respond to the rebuttal, despite the AC making repeated attempts to bug them. The AC reviewed the rebuttal and identified that the reviewer had highlighted three weaknesses: (1) the need for clarification regarding pitfalls in the generated summaries, (2) differences and challenges compared to related datasets, and (3) the need for a more advanced baseline for audio-visual fusion. The rebuttal appeared to address the first two clarification issues.

Although the third point about advanced baselines remains, the AC thinks that the strengths of the submission outweigh its weaknesses, particularly considering its contributions to the dataset track. Consequently, the AC recommends accepting the submission.

**Additional Comments On Reviewer Discussion:**

Two reviewers interacted with the authors, with one improving their score. See the details in the meta-review.

---

### Decision · Program_Chairs · 2025-01-22

Accept (Poster)